# GUI-Reflection: Empowering Multimodal GUI Models with Self-Reflection Behavior

Penghao Wu♠, Shengnan Ma♡, Bo Wang♡, Jiaheng Yu♡, Lewei Lu♡, Ziwei Liu♠ *

S-Lab, Nanyang Technological University♠, SenseTime Research♡

Project Page: `https://penghao-wu.github.io/GUI_Reflection/`

Figure 1: Illustrative comparison of typical GUI models versus our proposed GUI model with self-reflection behaviors. While current models fail to recognize and recover from errors (left), our model (right) demonstrates the ability to: 1. Recognize its mistake; 2. Undo the incorrect action and get back on track; 3. Summarize the mistake and make another try, ultimately succeeding.

## Abstract

Multimodal Large Language Models (MLLMs) have shown great potential in revolutionizing Graphical User Interface (GUI) automation. However, existing GUI models mostly rely on learning from nearly error-free offline trajectories, thus lacking reflection and error recovery capabilities. To bridge this gap, we propose **GUI-Reflection**, a novel framework that explicitly integrates self-reflection and error correction capabilities into end-to-end multimodal GUI models throughout dedicated training stages: *GUI-specific pre-training, offline supervised fine-tuning (SFT), and online reflection tuning*. GUI-reflection enables self-reflection behavior emergence with fully automated data generation and learning processes without requiring any human annotation. Specifically, **1)** we first propose scalable data pipelines to automatically construct reflection and error correction data from existing successful trajectories. While existing GUI models mainly focus on grounding and UI understanding ability, we propose the **GUI-Reflection Task Suite** to learn and evaluate reflection-oriented abilities explicitly. **2)** Furthermore, we built a diverse and efficient environment for online training and data collection of GUI models on mobile devices. **3)** We also present an iterative online reflection tuning algorithm leveraging the proposed environment, enabling the model to continuously enhance its reflection and error correction abilities. Our framework equips GUI agents with self-reflection and correction capabilities, paving the way for more robust, adaptable, and intelligent GUI automation, with all data, models, environments, and tools to be released publicly.

---

*Corresponding authors: `ziwei.liu@ntu.edu.sg`

39th Conference on Neural Information Processing Systems (NeurIPS 2025).

# 1 Introduction

Graphical User Interface (GUI) automation stands as a critical frontier for enhancing productivity and accessibility across the vast landscape of digital applications and devices. The advent of large language models (LLMs) and multimodal large language models (MLLMs) has catalyzed significant progress in this area. Typical GUI agents can be mainly categorized into two groups: agent-based frameworks and end-to-end GUI models. Agent-based frameworks [49, 52, 40, 39, 1] usually leverage the reasoning and generalization capabilities of some foundation models (*e.g.* GPT-4o [20]) with agentic modules and external tools to complete tasks. While representing a leap forward, such agent-based modular frameworks often rely on intricate prompt engineering and complex workflows, suffering from high computation and cascaded errors, potentially limiting their adaptability in real-world scenarios. As for end-to-end multimodal GUI models [14, 43, 44, 30] which interact with GUIs more like humans do, the perception, reasoning, and action grounding are integrated within a single model. Such models not only promise more adaptable and scalable GUI agents but also provide a potentially valuable avenue for studying broader aspects of artificial general intelligence [2].

The current paradigm for training end-to-end multimodal GUI models includes a GUI pre-training stage to inject GUI-related knowledge into the base MLLM, followed by a supervised fine-tuning (SFT) stage with demonstration trajectories. However, one big issue of this paradigm is that it relies heavily on offline datasets composed of pre-collected successful interaction trajectories. While this approach teaches models to mimic expert demonstrations for specific tasks, it inherently limits their ability to handle the complexities and unpredictability of real-world interactions. When encountering unfamiliar UI interfaces, incorrect attempts, or execution failures, these models lack the capability to recognize the error, understand its cause, or formulate a corrective plan based on failed attempts. Crucially, even though the base models (the base MLLM before GUI-specific training) they trained on might originally contain certain reflection and reasoning abilities, this offline SFT process, focused solely on successful examples, can inadvertently diminish such capabilities or behaviors.

Recent LLM research has shown that through online training like reinforcement learning (RL), the reasoning and reflection abilities of the base models can be greatly enhanced. Moreover, recent studies [34, 48, 15] have shown that the verification and reflection behaviors of the base model are crucial for the success of RL training, and such capabilities in the base models largely affect the performance upper bound in the RL stage. However, for end-to-end multimodal GUI models, the current paradigm makes it difficult to sample or explore potentially corrective or reflective behaviors after the offline SFT stage, and further RL training cannot effectively activate or enhance such reflection abilities. UI-TARS [30] explores the incorporation of reflection and correction behaviors through an online bootstrapping mechanism. However, its design primarily focuses on the final online stage and depends on human-annotated feedback to guide learning.

To address these fundamental limitations, we introduce an automatic framework designed to explicitly integrate self-reflection and error correction capabilities into end-to-end multimodal GUI models throughout different training stages. We first decompose the reflection and error correction ability for GUI agents into three core capabilities: (1) verifying the correctness of previous actions and recognizing errors or deviations, (2) backtracking when deviating from the correct trajectory, and (3) reflecting on erroneous attempts to learn from mistakes and inform subsequent actions. Our framework strategically embeds the learning of these abilities across distinct training phases, including GUI-specific pre-training, offline supervised fine-tuning, and online training, aiming to cultivate GUI models capable of robust error handling and adaptive recovery as illustrated in Fig 1.

Specifically, during the GUI pre-training phase, while current efforts primarily focus on GUI visual grounding and general UI understanding, we identify a critical gap: the lack of explicit training signals for reflection and correction related abilities, which leads to the degradation of reflection behaviors in the base MLLM. To address this, we propose the GUI-Reflection Task Suite with Action Verification, Action Reversal, and Mistake-Informed Reattempt tasks, specifically designed to evaluate and cultivate the reflection-oriented capabilities for GUI models. Besides, we have designed a scalable automatic data pipeline to construct realistic reflection and error correction scenarios derived from existing offline successful trajectories and inject such data into the offline SFT stage. This allows the model to learn the behaviors of reflection and correction.

---

[2] See Appendix A for a detailed discussion of related work.

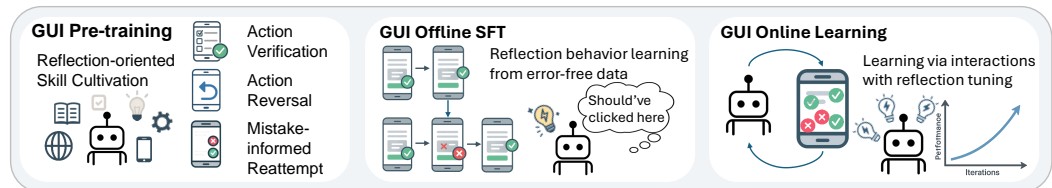

Figure 2: The GUI-Reflection framework includes (1) Learning basic reflection-oriented skills from GUI-Reflection Task Suite in the GUI pre-training stage; (2) Learning reflection and correction behaviours from automatically generated error scenarios in the offline SFT stage; (3) Continuously enhancing reflection and correction capabilities via reflection tuning in the online learning stage.

Furthermore, to further improve the GUI models through real interactions, we have built a robust and extensible environment for Android-based tasks. The environment includes 215 programmatic task templates in a distributed framework. Within this learning environment, we design a simple iterative online reflection tuning algorithm. This algorithm allows the model to interact with the environment and receive automatically generated pre-error correction and post-error reflection supervisions to continuously enhance its general capability and reflection and error correction capabilities iteratively.

Our main contributions are: **1)** We propose GUI-Reflection Task Suite, designed to explicitly train and evaluate the crucial reflection-oriented abilities of GUI models during GUI pre-training. **2)** We introduce a scalable, automatic data pipeline to construct reflection and error correction scenarios from existing successful trajectories, enabling the injection of these behaviors during offline SFT without manual annotation. **3)** We develop an online learning environment for mobile GUI agents and an iterative online reflection tuning algorithm, allowing models to continuously enhance their reflection and error correction capabilities through online interaction and learning from mistakes.

## 2 GUI-Reflection Framework

In this section, we first describe the core architecture of our GUI model and then elaborate on our GUI-Reflection framework (illustrated in Fig 2), which injects the self-reflection and correction behaviors into GUI models through pre-training, offline SFT, and online training stages.

### 2.1 End-to-End Multimodal GUI Agent Model

#### 2.1.1 Action Space

As we mainly focus on mobile tasks, the action space includes the following common atomic actions for mobile device interactions: `Click, Long Press, Scroll, Type, Press Enter, Press Back, Press Home, Open App, Task Impossible, Task Complete`. Besides the above common actions, we also include two actions that are often ignored in current datasets or methods. First, the GUI model often needs to retrieve and integrate information or answer certain questions, therefore, an `Answer` action is added into the action space for the agent to provide certain information or answers. Second, many tasks often require dozens of steps to complete, and it would be implausible and inefficient to contain complete history information like past screenshots in the context. Therefore, it is necessary to extract and keep certain information obtained in the process of the task execution as a reference for the usage in future steps (*e.g.* the model needs to search for different information online and send the information to someone via email). To achieve this, we additionally define a `Memorize` action which stores certain important information at certain steps into a memory bank for future references. The detailed descriptions of actions are provided in Appendix E.

#### 2.1.2 Model Structure

We adopt InternVL2.5-8B[13] as the base MLLM of our GUI model. The input of the model includes the following parts: 1) The overall instruction of the task; 2) The screenshots of the past $n$ steps; 3) The screenshot of the current step; 4) The memory bank; 5) The complete action history.

The model output consists of three parts: 1) action thought, 2) action description, and 3) atomic action. The action thought is the thinking process behind the action decided to take. And it may

contain aspects including analysis of the current screen state, assessment of the prior step's outcome, reflection about previous steps' actions, consideration of overall task progress, and the rationale for the chosen action. These elements are included dynamically and cohesively, only when pertinent to the decision process, without rigid structure or explicit labels. The action description describes the action in natural language, while the action atomic is a certain action type with the corresponding parameters. Detailed input and output formats and examples can be found in Appendix E.

Based on the input and output definitions, a training sample at step $t$ in a trajectory can then be defined as $(G, M_t, I_{t-n:t}, a_{0:t-1}; a_t^{thought}, a_t^{desc}, a_t)$, where $G$ denotes the overall task goal, $M_t$ denotes the current memory content at step $t$, $I_{t-n:t}$ represents the screenshots of the past $n$ steps and the current step, $a_{0:t-1}$ represents the complete past actions, $a_t^{thought}$ represents the action thought of the current step, $a_t^{desc}$ represents the action description of the current step, and $a_t$ is the grounded atomic action. We use $\boldsymbol{a}_t = (a_t^{thought}, a_t^{desc}, a_t)$ to represent the action outputs.

## 2.2 GUI-Reflection Task Suite: Reflection-oriented Abilities in Pre-training

While GUI grounding and understanding are crucial for basic GUI interactions, we argue that it is also important to maintain or enhance the model's nascent abilities for self-reflection and error recognition within the GUI context. In this pre-training stage, we do not directly incorporate the complete GUI-related reflection and correction behaviors, instead, we further decompose such reflection and correction behaviors into smaller reflection-oriented atomic capabilities and design the GUI-Reflection Task Suite to evaluate and learn such capabilities.

**Action Verification** A GUI agent with reflection ability might execute an incorrect action due to limited knowledge or unfamiliarity with the task, but it would recognize the mistake by observing the outcome of the action. Recognizing the error or mistake is the very first and crucial step in the reflection and correction process. To address this foundational capability, our first introduced pre-training task is Action Verification. The core idea is to test the model's ability to determine if an implicit action, executed on a previous GUI state, accomplished a specific purpose, based on observing the resulting GUI state outcome.

In this task, the model is presented with screenshots of two consecutive steps together with a textual action purpose describing potential goals or outcomes that the action performed on the first screen aimed to achieve. The model's objective is to meticulously inspect the visual differences between the screenshots and judge whether that specific purpose was successfully fulfilled by the implicit action. To construct data for this task, we randomly sample paired screenshots together with the corresponding action from GUI trajectory datasets. We adopt an MLLM to annotate the true action purpose for this action to be a positive sample and annotate a negative purpose, which corresponds to a different action and is not accomplished in the second outcome screenshot.

**Action Reversal** Our second task is termed Action Reversal. This task addresses the scenario where an undesired or incorrect action has been recognized, and the objective is to determine the subsequent action required to revert the GUI to its state immediately preceding the execution of the original action. In essence, the model learns how to effectively undo a given action and eliminate its consequences. This capability is crucial for enabling more sophisticated error correction and exploration strategies.

We define this task as multiple-choice questions. The step-wise action, paired with the screenshot before and after the action execution, is presented to the model, and the model needs to choose the correct undo action. We construct the data from GUI trajectory datasets and use MLLMs to annotate the undo action and interference options.

**Mistake-informed Reattempt** After recognizing an error and potentially reverting the state, a critical reflective capability is to make an informed new attempt based on the known mistakes. To evaluate this ability, we introduce the Mistake-informed Reattempt task. In this task, the model is first asked to ground GUI elements based on a given instruction. We then identify the samples that are incorrectly grounded. The model is informed of the prior mistake and is asked to make a new prediction. This process can also be repeated multiple times with multiple failed attempts.

We construct both training and evaluation data for these three tasks. For the Action Verification and Action Reversal tasks, the training and evaluation data are constructed from the training and test splits of AndroidControl [21] and GUI-Odyssey [27]. We have 1206 and 420 samples for the evaluation of these two tasks, respectively, where the evaluation data has been filtered by human annotators to

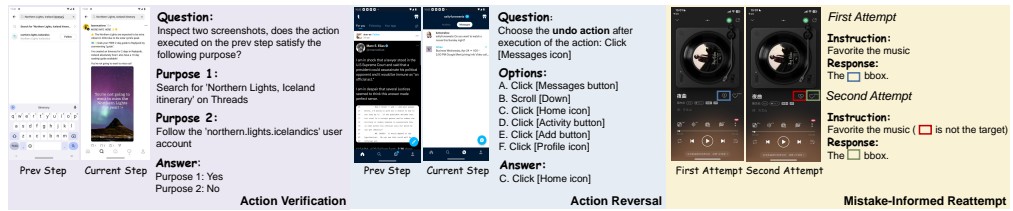

Figure 3: Examples of Action Verification (left), Action Reversal (middle), and Mistake-informed Reattempt (right) tasks from the GUI-Reflection Task Suite.

ensure correctness. For the Mistake-informed reattempt task, the training data is constructed from Wave-UI [2], AMEX [11], and OS-ATLAS-Desktop [43]. For this task, we evaluate directly on ScreenSpot [14] and ScreenSpotv2 [43]. Examples of these three tasks are provided in Fig 3. The detailed data construction process and statistics are provided in Appendix F.

## 2.3 Automatic Grounded Action Annotation

Before introducing the reflection data generation for the offline SFT stage, we first discuss how to generate grounded action annotations for GUI models without human annotations. End-to-end GUI models need to provide the final grounded action (*e.g.* the start and end position of scroll action or the clicked point of click action). Therefore, one difficulty of automatic action annotation for end-to-end GUI models is to generate the grounded action together with the paired action thought consistently. In order to solve this problem, we utilize the strong generalization ability of the current general MLLMs and the specific action grounding ability of GUI models. More precisely, to annotate the action outputs (including the action thought, action description, and the grounded atomic action), we first utilize general MLLMs to generate the desired action thought and action description. Next, we concatenate the generated action thought and description together with the input information for the GUI model, and let the GUI model output the grounded atomic action correspondingly. Due to the auto-regressive nature of LLM, the GUI model outputs the atomic action conditioned on the provided action thought and action description. To ensure the generated grounded atomic action is consistent with the action thought and action description, we sample multiple atomic actions from the GUI model and utilize the MLLMs again for filtering.

## 2.4 Reflection Behavior in Offline SFT

During the SFT stage, the GUI model is trained on offline GUI interaction trajectories that are mostly error-free. The ability to recognize possible mistakes based on execution results and the ability to recover or learn from mistakes are greatly limited in such a training approach. Therefore, we design a scalable automatic data pipeline to create realistic reflection and correction data from the existing successful trajectories.

The difficulty of creating reflection and correction data from existing error-free trajectories is how to get an incorrect action and its corresponding outcome screenshot. We design two approaches to address this problem. For the first approach, we adopt an MLLM to modify the original goal $G$ to $\tilde{G}$ such that an action $a_t$ at a certain step $t$ becomes an incorrect one. The modified goal is constructed to make the now-incorrect action appear as an easy or natural mistake that a user unfamiliar with the app, button functions, or certain operations might make. Based on $\tilde{G}$ and $a_t$, we now construct the new action outputs at step $t + 1$ after the mistake. At step $t + 1$, the agents should recognize the previous mistake made in the last step and make reflections in $\tilde{a}_{t+1}^{thought}$, with $\tilde{a}_{t+1}^{desc}$ and $\tilde{a}_{t+1}$ generated accordingly. Note that $\tilde{a}_{t+1}$ could be some rollback action, such as `press back`, if the previous incorrect action $\tilde{a}_t$ leads to an off-track state or could also be some correction action where the agents can directly continue with the correct action towards the modified goal. A reflection data sample $(\tilde{G}, M_{t+1}, I_{t-(n-1):t+1}, a_{0:t}; \tilde{a}_{t+1}^{thought}, \tilde{a}_{t+1}^{desc}, \tilde{a}_{t+1})$ is then constructed.

Furthermore, for cases where $\tilde{a}_{t+1}$ is `press back`, we assume the screenshot $\tilde{I}_{t+2}$ after execution of $\tilde{a}_{t+1}$ is the same as $I_t$. Then we can further generate action output for step $t + 2$, in which we summarize the previous error in step $t$, make reflections, and try a new correct action $\tilde{a}_{t+2}$. This simulates the scenario where the agent learns from its mistake after backtracking and attempts an

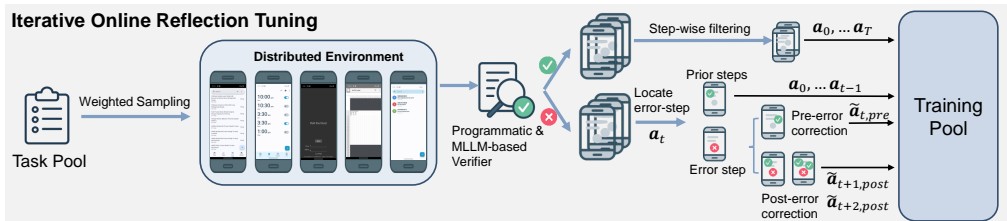

Figure 4: The Iterative Online Reflection Tuning algorithm. It features weighted task sampling, interaction in a distributed environment, and programmatic/MLLM-based verification. Successful trajectories are further filtered step-wise, while unsuccessful ones are mined for correct prior steps and undergo automated pre-error and post-error (reflection) correction annotation.

alternative action to achieve the goal from the restored state. And this data sample can be represented as $(\tilde{G}, M_{t+1}, [I_{t-(n-2):t+1}, I_t], [a_{0:t}, \tilde{a}_{t+1}]; \tilde{a}_{t+2}^{thought}, \tilde{a}_{t+2}^{desc}, \tilde{a}_{t+2})$.

For the second approach, we keep the original task instruction goal. For some step $t$ in the original successful trajectory, we construct an ineffective incorrect action $\tilde{a}$ which should not change the screenshot $I_t$ (*e.g.* scroll down when it is already the bottom or click on some non-interactive element). Then we assert this action before the actual $a_t$, and modify the original $a_t^{thought}$ to $\tilde{a}_t^{thought}$ by adding reflection content about the added ineffective incorrect action. The data sample created in this approach is represented as $(G, M_t, [I_{t-(n-1):t}, I_t], [a_{0:t-1}, \tilde{a}]; \tilde{a}_t^{thought}, a_t^{desc}, a_t)$.

## 2.5 Iterative Online Reflection Tuning

### 2.5.1 Environment

Effective online training necessitates a diverse, efficient, and scalable environment. However, current public online environments for training mobile GUI agents [7, 41] only include overly simple or repetitive tasks, which lack the complexity and diversity agents likely to encounter in real-world scenarios. To overcome these limitations, we developed a specialized environment for efficient online learning, testing, and data collection of mobile GUI agents.

Specifically, our environment includes 215 task templates across 11 Apps. Each task template can be instantiated randomly with dynamic parameters. Based on the complexity of the tasks, we split the tasks into two levels, where 135 level-1 tasks include relatively easier ones, and 80 level-2 tasks have higher complexity. Our platform is a distributed host-worker system. The workers only run CPU-intensive Android Emulators, and all GPU-related inference or training tasks are running on the host machine. For trajectory collection, the agent model deployed on the host machine receives environment observations from the worker and sends the predicted action back to the worker to interact with the environment.

For task evaluation, we support both programmatic and MLLM-based verifiers. Programmatic verifier directly evaluates the success of the task by accessing the device's system states and databases, providing accurate reward signals. For the MLLM-based verifier, the task information, action history, and the corresponding screenshots will be provided to an MLLM, which determines whether the task is successful. This is helpful for tasks where some critical information or intermediate results needed for the evaluation are not accessible from the device states. Besides, the MLLM-based evaluation is also able to check the step-wise correctness of the trajectory, providing dense process reward. To improve the accuracy of the MLLM-based evaluations, we also provide task guidance for each task template, describing the general procedures and important points of the task. Details about the environment and verifiers are provided in Appendix H.

### 2.5.2 Algorithm

We design an iterative reflection tuning algorithm for the GUI model trained with offline SFT to further improve the general and reflection capabilities through interacting with our online environment.

To elaborate, in each iteration, the current GUI model collects multiple rollouts by interacting with the environment. Different from the regular filtered behavior cloning algorithm, which directly takes all successful trajectories for training and discards the unsuccessful ones, for successful trajectories,

Table 1: Evaluation results on the three tasks in GUI-Reflection Task Suite

(a) Evaluation results on Action Verification and Action Reversal tasks. Acc denotes the overall accuracy. TN and TP denote the accuracy for negative samples (action purpose not fulfilled) and positive samples (action purpose fulfilled), respectively. All numbers are in %. The best numbers among open-source models are in bold.

| Models | Action Verification | | | Action Reversal |
|---|---|---|---|---|
| | Acc | TN | TP | Acc |
| Gemini-2.5-Flash | 87.85 | 92.02 | 83.69 | 95.24 |
| Gemini-2.5-Pro | 88.22 | 88.04 | 88.40 | 95.71 |
| Claude-3.7-Sonnet | 71.53 | 57.48 | 85.58 | 90.00 |
| GPT-4o | 86.68 | 93.11 | 80.25 | 86.19 |
| Qwen2.5-VL-7B | 76.36 | 69.32 | 83.41 | 76.90 |
| Qwen2.5-VL-72B | 86.48 | 90.38 | 82.59 | 91.90 |
| InternVL2.5-8B | 62.76 | 51.07 | 74.46 | 48.33 |
| InternVL3-8B | 60.11 | 26.86 | 93.36 | 63.80 |
| InternVL3-78B | 68.24 | 52.07 | 84.41 | 82.38 |
| GUI-Pretrain-8B | 57.95 | 21.55 | 94.36 | 40.71 |
| GUI-Pretrain-Ref-8B | **87.56** | 93.53 | 81.59 | **93.81** |

(b) Evaluation results on the Mistake-informed Reattempt task. The average scores of the mobile, desktop, and website subsets of the benchmarks are reported. The best numbers are in bold.

| | ScreenSpot | ScreenSpotv2 |
|---|---|---|
| InternVL3-8B | 71.59 | 72.02 |
| - 2nd attempt | 73.84 (↑ 2.25) | 74.42 (↑ 2.40) |
| - 3rd attempt | 75.13 (↑ 3.54) | 75.73 (↑ 3.71) |
| - pass@3 | 77.90 (↑ 6.31) | 80.09 (↑ 8.07) |
| GUI-Pretrain | 83.58 | 84.84 |
| - 2nd attempt | 84.50(↑ 0.92) | 85.65 (↑ 0.81) |
| - 3rd attempt | 84.75(↑ 1.17) | 85.85 (↑ 1.01) |
| - pass@3 | 87.24 (↑ 3.66) | 88.39 (↑ 3.45) |
| GUI-Pretrain-Ref | 85.12 | 86.88 |
| - 2nd attempt | 88.00 (↑ 2.88) | 89.27 (↑ 2.61) |
| - 3rd attempt | **89.61** (↑ 4.49) | **90.50** (↑ 3.62) |
| - pass@3 | 87.35 (↑ 2.23) | 88.84 (↑ 1.96) |

we additionally check the step-wise correctness of each step and only keep the correct steps. For unsuccessful trajectories, we find the first step $t$ where the model performs an incorrect action. And all the steps before $t$ are kept for training. Then, for the incorrect action $a_t$ at step $t$, a pre-error correction action $\tilde{a}_{t,pre}$ is annotated to be the actual correct action for step $t$. For the step $t+1$ after the execution of the incorrect action $a_t$, a post-error correction action $\tilde{a}_{t+1,post}$ is annotated, in which the model recognizes the previous mistake and makes reflections. Furthermore, when the action $\tilde{a}_{t+1,post}$ is `Press Back`, we assume the screenshot $\tilde{I}_{t+2}$ is the same as $I_t$ and assign $\tilde{a}_{t+2}$ to be the same action as $\tilde{a}_t$, with additional reflection added to the action thought summarizing the previous mistakes and making a new try accordingly. All the action annotations are automatically constructed utilizing a general MLLM and the current GUI model. The training data collected in this iteration is used to fine-tune the current GUI agent model. This process is illustrated in Fig 4.

After each iteration, the sampling weights for different types of tasks are dynamically adjusted based on the corresponding success rates in this iteration, such that more difficult tasks are sampled more in the next iteration. We also adopt the curriculum learning strategy, where in the first $k$ iterations, only level-1 tasks are included. After the first $k$ iterations, level-1 tasks with success rates lower than a certain threshold are kept and combined with level-2 tasks for subsequent iterations.

## 3 Experiments

### 3.1 Training Data

For the GUI pre-training stage, besides the reflection-related data constructed, we also include GUI-related grounding, captioning, OCR, and VQA data. For the GUI offline SFT stage, we use public mobile device GUI interaction datasets including AITW [33], AITZ [50], AMEX [11], GUI-Odyssey [27], and AndroidControl [21]. We unify the action annotation of these datasets and annotate the corresponding action thought and action description via Gemini-2.0-Flash [38]. The detailed statistics of the training data are provided in Appendix D.

### 3.2 Evaluations on GUI-Reflection Task Suite

In this section, we evaluate tasks in our GUI-reflection Task Suite. For the Action Verification task and Action Reversal task, we include 1) closed-source models: Gemini-2.5-Flash/Pro [38], Claude-3.7-Sonnet [4], and GPT-4o [20]; 2) open-source models: Qwen2.5-VL-7/72B [9], InternVL2.5-8B [13], and InternVL3-8/78B [53]; 3) GUI baseline: GUI-Pretrain, which is our base MLLM pre-trained with regular GUI pre-training data, and GUI-Pretrain-Ref, which is pre-trained with additional GUI-Reflection Task Suite training data.

As shown in Table 1a, powerful closed-source models perform strongly on these two tasks. The 72B scale open-source models perform comparatively well, while 7B scale models have lower performance, especially for the capability to recognize the failed action (TN in Action Verification) that is critical for recognizing mistakes in the reflection process. Note that after the regular GUI pre-training, the GUI-Pretrain model performs much worse than the previous general MLLM, indicating the loss of such reflection-related abilities after GUI-specific pre-training. After adding the corresponding data in the GUI pre-training phase, GUI-Pretrain-Ref retains and even greatly improves such capabilities, performing on par with the best closed-source models.

For the Mistake-informed Reattempt task, we evaluate models' ability to reattempt based on known mistakes on the instruction grounding benchmarks ScreenSpot [14] and ScreenSpotv2 [43]. For samples that are incorrectly grounded, we provide the incorrect predictions and ask the model to make another attempt accordingly, and repeat this process for 2 rounds. We also provide the pass@3 (temperature=1.0) results for comparison. As shown in Table 1b, we observe that the general MLLM InternVL3-8B and the GUI pre-trained model GUI-Pretrain pose limited ability to utilize the known mistakes for more informed attempts (the 3rd attempt performance is lower than pass@3). After adding our reflection-related training data, the GUI-Pretrain-Ref baseline can more effectively utilize the mistakes to make better predictions (the 3rd attempt performance is higher than pass@3).

These experimental results show that large-scale general-purpose MLLMs possess some inherent reflection capabilities in the GUI context, while such capabilities are still very limited in smaller-scale models, and the standard GUI pre-training tends to further diminish these abilities. However, by incorporating training data from our reflection-oriented tasks during the pre-training phase, such essential capabilities can be effectively improved.

### 3.3 Effectiveness of Reflection for GUI Agents

In this part, we continue from the GUI-Pretrain-Ref model and conduct experiments to validate the effectiveness of reflection data in the SFT and online stages. First, we verify the effect of augmenting offline GUI SFT data with reflection data and conducting iterative reflection tuning in the online environment. We conduct experiments in our GUI environment by training models with the level-1 tasks for 3 iterations and evaluating the performance on the level-2 tasks. As shown in Table 2, the baseline model trained without reflection data in offline SFT and using only filtered BC achieves a success rate of 14.58% on level-2 tasks. Incorporating reflection data during the offline SFT stage significantly boosts this to 23.61% with the same filtered BC online training. Critically, when our online reflection tuning algorithm is applied online, the success rate further improves to 34.72%, demonstrating the benefits of explicitly training for reflection at multiple stages.

Table 2: Ablation study on reflection data in SFT and online training.

| Reflection SFT | Online Algo. | Success (%) |
|:---:|:---:|:---:|
| ✘ | Filtered BC | 14.58 |
| ✔ | Filtered BC | 23.61 |
| ✔ | + Reflection Tuning | **34.72** |

Table 3: Comparison of our model against other baselines on AndroidWorld, showing Success Rates (SR). Acc. Tree denotes Accessibility Tree. The best number in 8B-scale end-to-end models is marked in bold.

| Baseline | Input | SR |
|---|:---:|:---:|
| *Agent-Based* | | |
| GPT-4o + UGround [16] | Image + Acc. Tree | 32.8 |
| GPT-4o + Aria-UI [45] | Image + Acc. Tree | 44.8 |
| GPT-4o + Aguvis-7B [44] | Image | 37.1 |
| *End-to-End* | | |
| Aguvis-72B [44] | Image | 26.1 |
| UI-TARS-72B [30] | Image | 46.6 |
| OS-Gensis-8B [36] | Image + Acc. Tree | 16.9 |
| UI-TARS-7B [30] | Image | 33.0 |
| **GUI-Reflection-8B** (Ours) | Image | **34.5** |

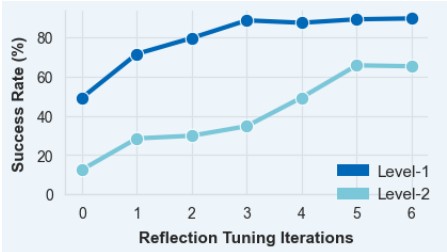

Figure 5: Success Rate (%) on Level-1 and Level-2 tasks across iterative reflection tuning iterations. Our iterative reflection tuning with curriculum learning strategy progressively improves model performance.

We further conduct the full reflection tuning process in the online environment with all tasks. In the first 3 iterations, only level-1 tasks are included, and the tasks with success rates lower than 80% are combined with level-2 tasks for another 3 iterations. As shown in Fig 5, the model rapidly improves on level-1 tasks, eventually reaching around 90% and maintaining this high performance. For the more complex level-2 tasks, the agent starts at a lower success rate and shows steady improvement, reaching 29.36% by iteration 3. When more challenging level-1 tasks are combined with all level-2 tasks, the learning on level-2 tasks continues robustly. This illustrates the effectiveness of the online reflection tuning algorithm, enabling the model to effectively enhance its general ability and rapidly learn to master previously unfamiliar and complex tasks.

Furthermore, to evaluate our model on more general and comprehensive tasks, we combine the training data collected in the online training stage with a similar-sized subset of the original offline data and fine-tune the offline SFT model to inject valuable reflection experiences while maintaining the generalization ability. We evaluate our model on the AndroidWorld [32] benchmark. As shown in Table 3, our model achieves a competitive success rate of 34.5% among end-to-end models, demonstrating the effectiveness of our proposed framework.

**Task Instruction**
Toggle on all alarms that are in AM in the Fossify Clock App.

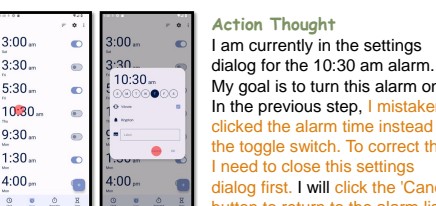

**Action Thought**
I am currently in the settings dialog for the 10:30 am alarm. My goal is to turn this alarm on. In the previous step, I mistakenly clicked the alarm time instead of the toggle switch. To correct this, I need to close this settings dialog first. I will click the 'Cancel' button to return to the alarm list.

**Task Instruction**
In the LibreOffice Viewer App, open the njrh7.docx under the Meetings folder in the sdk_gphone64_x86_64 storage area. Then export it to a PDF with name er00f.pdf in the same director.

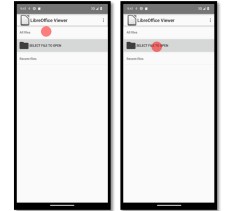

**Action Thought**
I see that clicking 'All files' in the previous step did not change the screen, indicating it was an incorrect action. I mistakenly thought clicking the section header would reveal browsing options, but it seems it's just a label. To proceed with opening the specified file, I need to initiate the file selection process. The correct way to do this is to click the 'SELECT FILE TO OPEN' button.

**Task Instruction**
In Simple Calendar Pro, create a calendar event in two weeks from today at 17h with the title 'Meeting with the Team' and the description 'We will prepare for annual budget.'. The event should last for 30 mins.

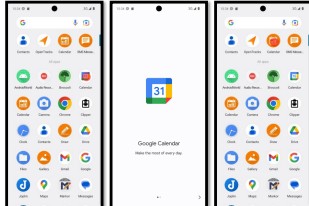

**Action Thought – Step2**
I intended to open the Simple Calendar Pro app, but I mistakenly clicked the Google Calendar app icon. Now I am in the Google Calendar app, which is not the correct app for this task. I need to go back to the app drawer to find and open the correct Simple Calendar Pro app.

**Action Thought – Step3**
I am currently in the app drawer. I previously clicked the wrong Calendar app (the blue one with '15'), realizing it wasn't the required Simple Calendar Pro. I need to find the correct app to proceed with creating the calendar event. I see two calendar apps here; the correct one is the orange icon with the text 'Calendar' below it. I will click this icon now.

Figure 6: Illustrative examples of our GUI model's self-reflection and correction behaviours. The model demonstrates its ability to: (Top-left) identify and undo a mis-click; (Top-right) recognize an unproductive action and then find the correct interactive element; and (Bottom) recover from opening an incorrect app by navigating back and selecting the correct one based on learned cues.

**Reflection Behavior Examples** We also provide examples of reflection and correction behaviors from the GUI models trained with our framework in Fig 6. We observe that, without our explicit reflection training, GUI models typically struggle to recognize or recover from errors, often leading to task failure when encountering unexpected situations. Conversely, our framework enables models to analyze mistakes and execute corrective steps, effectively navigating towards the task goal despite initial incorrect attempts.

## 4   Conclusion

This paper introduced GUI-Reflection, a comprehensive framework designed to equip multimodal GUI models with essential self-reflection and error correction capabilities. By systematically integrating reflection-related learning across pre-training, offline SFT, and online tuning, GUI-Reflection enables agents to recognize their mistakes, undo incorrect actions, and learn from these errors to make better subsequent decisions.

## Acknowledgments

This study is supported by the Ministry of Education, Singapore, under its MOE AcRF Tier 2 (MOE-T2EP20221-0012, MOE-T2EP20223-0002), and under the RIE2020 Industry Alignment Fund – Industry Collaboration Projects (IAF-ICP) Funding Initiative, as well as cash and in-kind contribution from the industry partner(s).

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

# A    Related Works

## A.1    Mobile GUI Agents

Driven by the success of large language models and multimodal large language models, research in mobile GUI automation has seen significant advancements. Current approaches to developing mobile GUI agents can be broadly categorized into agent-based frameworks and end-to-end models, each with distinct characteristics and trade-offs.

**Agent-based** By leveraging the advanced reasoning, planning, generalization capabilities, and broad knowledge of foundation LLMs (*e.g.* GPT-4o), agent-based frameworks structure sophisticated agentic workflows. One approach uses foundation models to directly engage with GUI interfaces [49, 22, 39]. Such methods usually depend on accessible device information, particularly accessibility trees, to allow the model to ground actions at the element level. These systems primarily consume textual information from accessibility trees, often enriched with screenshots featuring Set-of-Mark (SoM) augmentations for improved visual understanding. To enhance their operational efficacy, these workflows frequently incorporate components such as memory [39], reflection [22, 39], knowledge documents [49, 22], task decomposition [39], and visual tool integration [28], thereby improving task completion and overall agent robustness.

Another agent-based approach [16, 45] combines a powerful foundation model with a specialized GUI grounding model. In this setup, the LLM handles high-level planning and reasoning, while the dedicated GUI grounding model is responsible for accurately identifying and interacting with GUI elements based on a low-level instruction or element description from the LLM. UGround [16] trains a universal GUI grounding model and combines it with a planning model. Aria-UI [45] improves the grounding model by providing more task context, like overall task instruction and history information.

**End-to-end** End-to-end GUI models [18, 14, 44, 43, 51, 30] aim to directly map raw GUI inputs (task information and screenshots) to grounded actions within a single model. The common paradigm for these models involves a two-stage training process: 1) GUI-specific pre-training , where the model learns fundamental GUI understanding and accurate grounding ; 2) GUI offline SFT , where the pre-trained model is fine-tuned on demonstration trajectories to learn task-specific behaviors . Some methods [7, 41, 8] also apply reinforcement learning on the fine-tuned model, but experiments are only conducted in relatively simple and repetitive tasks. Beyond only predicting the atomic action, recent methods adopt the CoT idea to train the model to output additional components like the thinking process [50, 44, 30] and low-level action description [50, 44]. InfiGUIAgent [24] adds expectation-reflection components in the training data, but they only use the existing offline data, with most steps being error-free, and the reflection component primarily learns to confirm success rather than to actively diagnose and recover from a wide array of potential failures. UI-TARS [30] introduces an online bootstrapping process to learn the self-reflection and correction behaviors. However, unlike our proposed framework, which employs a fully automated pipeline for generating reflection data and integrates reflection capability enhancement across pre-training, offline SFT, and online stages, UI-TARS's approach requires considerable human annotation efforts and focuses the learning of these behaviors only in their final online bootstrapping process.

## A.2    LLM and MLLM Reasoning and Reflection

The pursuit of enhanced reasoning in Large Language Models (LLMs) has evolved from structured prompting and SFT data construction [42, 46, 47] towards leveraging Reinforcement Learning. While initial RLHF methods [35, 31] showed promise, recent paradigms focusing on outcome-based rewards have demonstrated great potential to intrinsically cultivate complex reasoning and even emergent self-reflection [17]. For the multimodal domains, current research is actively exploring how to adapt similar RL techniques to improve multimodal reasoning [19, 25, 29] involving visual information like images and videos, though challenges related to data and effective training signals remain. Furthermore, recent studies underscore the critical importance of the inherent capabilities and behaviors present in the base models before task-specific fine-tuning or reinforcement learning begins. Research indicates that foundational abilities for verification and reflection are not merely helpful but often prerequisites for successful online learning and significantly influence the ultimate performance ceiling achievable through RL [34, 48, 15]. This highlights a potential vulnerability in current end-to-end GUI model training pipelines, which often rely heavily on offline SFT with near

error-free data. Such approaches may inadvertently suppress or fail to cultivate these vital reflective capabilities present in the base MLLM.

## B   Limitation

In this work, the constructed reflection-related data focuses primarily on visual and action-grounded errors or direct element functioning misunderstanding, potentially neglecting deeper and more complex errors, such as errors in high-level planning or complex task decomposition. Besides, our framework currently mainly focuses on mobile environments. While the underlying principles are generalizable, adapting GUI-Reflection to other platforms such as desktop systems or web-based interfaces may require domain-specific dataset construction and engineering adjustments.

## C   Societal Impacts

GUI-Reflection has the potential to improve digital accessibility and productivity by enabling more robust and error-tolerant GUI agents. However, these capabilities could also be misused for automated manipulation in malicious contexts. Furthermore, the reliance on synthetic data may introduce biases if not carefully curated, potentially leading to unintended behaviors in sensitive applications. Responsible deployment, transparency in usage, and alignment with human intentions are critical for maximizing societal benefit while minimizing risks.

## D   Training Details

Besides the reflection-related data we construct in the GUI-Reflection pipeline, the statistics of other data and the corresponding license information we used in the GUI pre-training and offline SFT stages are provided in Table 4 and Table 5.

Table 4: The detailed training data information for the GUI-Pretraining Stage. The total number reported is at the element level, while in implementation, the elements on the same image are grouped as a single training sample in the multi-turn conversation format.

| Data Source | Platform | Task Type | Total Samples | License |
|---|---|---|---|---|
| UI RefExp [6] | Mobile | Grounding | 16,660 | CC BY 4.0 |
| Widget Captioning [23] | Mobile | Grounding | 96,648 | CC BY 4.0 |
| SeeClick-Rico [14] | Mobile | Grounding | 173,275 | CC BY 4.0 |
| RICO Semantics [37] | Mobile | Grounding | 31,560 | CC BY-SA 4.0 |
| OpenApp [10] | Mobile | Grounding | 142,810 | BSD-3-Clause license |
| AMEX [11] | Mobile | Grounding | 1,360,595 | CC BY 4.0 |
| OS-Altas [43] | Mobile | Grounding | 89,860 | Apache-2.0 |
| Wave-UI [2] | Web | Grounding | 79,412 | MIT |
| Wave-UI-25K [3] | Web | Grounding | 24,978 | MIT |
| SeeClick-Web [14] | Web | Grounding | 2,968,695 | Apache-2.0 |
| GUIEnv [12] | Web | Grounding | 340,477 | CC BY 4.0 |
| ScreenQA [5] | Mobile | VQA | 62,401 | CC BY 4.0 |

Table 5: The detailed training data information for the offline SFT stage.

| Data Source | Platform | Total Steps | License |
|---|---|---|---|
| AITW [33] | Mobile | 19,831 | CC BY 4.0 |
| AITZ [50] | Mobile | 14,686 | CC BY 4.0 |
| AMEX [11] | Mobile | 39,023 | CC BY 4.0 |
| AndroidControl [21] | Mobile | 89,603 | Apache-2.0 |
| GUI-Odyssey [27] | Mobile | 102,202 | CC BY 4.0 |

For the GUI pre-training, we train the model for 1 epoch with a learning rate of $4 \times 10^{-5}$. For the SFT stage, we train the pre-trained model for 1 epoch with a learning rate of $3 \times 10^{-5}$. In each

reflection tuning iteration, we train the model on the collected data in this iteration for 2 epochs with a learning rate of $1 \times 10^{-5}$. For our final model, we randomly sample 51694 samples from the offline SFT data and combine them with 63353 samples collected in the online iterations and finetune the model after offline SFT for 1 epoch with a learning rate $2 \times 10^{-5}$. We use AdamW [26] optimizer for all the training. All the training is conducted on 32 H100 GPUs. The pre-training stage takes about 11.5 hours, and the SFT stage takes about 8.5 hours. Each training iteration during the online training stage needs about 2 hours, and the final training stage takes 4 hours.

# E  Implementation Details

## E.1  Model Details

The detailed descriptions of the valid actions for our GUI model are provided below.

```
Action Space

CLICK[[x, y]]. Click the screen at position [x,y].
LONG_PRESS[[x, y]]. Long press the screen at position [x, y].
SCROLL[[x1, y1, x2, y2]]. Scroll from the position [x1, y1] to [x2, y2].
TYPE[text]. Type in the text.
MEMORIZE[summary: text; content: text]. Store information into the memory.
ANSWER[text]. Answer with the text.
PRESS_HOME. Go back to the home screen.
PRESS_BACK. Go back to the previous screen.
OPEN_APP[app_name]. Open the app named app_name.
PRESS_ENTER. Press the enter key.
WAIT. Wait for device response.
TASK_COMPLETE. Indicate the task is completed.
TASK_IMPOSSIBLE. Indicate the task is impossible.
```

The input and output formats of our GUI agent model are shown below.

```
Input Format of the GUI Agent

<image>
<image>
The images are the screenshots from the past 2 steps.
<image>
The image is the current screenshot.
<INSTRUCTION> (user instruction): {goal}
<MEMORY> (stored memory content): {current memory}
<PAST ACTIONS> (past actions): {action history}
Based on the above information, your task is to reason about the next action
and provide your thinking process and the next action. Your output should
follow the following format:
<THOUGHT>: the thinking process
<ACTION DESC>: the description about the next action
<ACTION>: the next action
```

```
Output Format of the GUI Agent

<THOUGHT>: {action thought}
<ACTION DESC>: {action description}
<ACTION>: {action}
```

We define the action descriptions to follow fixed formats, and the formats for different action types are shown below.

```
Action Description Format

CLICK: click the {element} to {purpose}
LONG_PRESS: long press the {element} to {purpose}
SCROLL: scroll {direction} to {purpose}
TYPE: type in the content '{content}'
MEMORIZE: memorize {memory_summary}
ANSWER: answer with the text '{}'
PRESS_HOME. Go back to the home screen
PRESS_BACK. Go back to the previous screen
OPEN_APP: open the '{app_name}' app
PRESS_ENTER: press enter
WAIT: wait
TASK_COMPLETE: task complete
TASK_IMPOSSIBLE: task impossible
```

In our model implementation, the screenshot history length $n$ is set to 4. All past screenshots except the one in the last step are downsampled to $448 \times 448$. We also visualized the click point on the past screenshot using a red dot if the corresponding action is a click or long press. The coordinates in the action representations are normalized to integers in the range 0 to 999.

## E.2 Evaluation Details

The original scroll and swipe implementation in AndroidWorld [32] always uses a fixed trajectory, so we modify it to make the scroll action trajectory follow the start and end points predicted by the model. The original type action implementation includes clicking the target element and typing, so we modify it to only include typing the text, and the model needs two actions (click + type) to complete the original type action. We find that in some cases of AndroidWorld, the maximum steps defined are impossible for agents that do not use the UI element information, so we increase the maximum step by 5 for all test cases.

## F  Details of GUI-Reflection Task Suite

### F.1  Action Verification

For the action verification task, we randomly sample step-wise data from AndroidControl [21] and Odyssey [27] datasets. Each data sample consists of a ground truth action, the screenshot before the action, and the screenshot after the action. We only consider action types including CLICK, LONG PRESS, and SCROLL as the purposes of other actions are relatively fixed. Then, we extract the purpose from the action descriptions of the ground truth actions to be the positive purpose. To construct the negative purpose, we use Gemini-2.5-Pro to annotate the corresponding negative purpose for this sample. The prompt for this annotation is shown in Table 8.

For the evaluation data of the action verification task, we have 603 positive samples and 603 corresponding negative samples from the Odyssey test split. For the training data, we construct 16220 paired samples from AndroidControl and 15616 paired samples from Odyssey. The evaluation and training template for this task is shown in Table 9.

### F.2  Action Reversal

The data for this task is constructed in two steps. First, we sample step-wise action data paired with the screenshot before and after the action execution from existing datasets. Gemini-2.5-Pro is instructed to generate the appropriate *undo* action for this data pair as the ground truth. Our model is supposed to learn from this action reversal process. The annotation MLLM is instructed to prioritize app-internal revert actions instead of overly relying on the general Press Back action during this *undo* action generation process. When multiple revert actions are available, we select the most straightforward and efficient option. After obtaining the correct *undo* action, we instruct Gemini-2.0-Flash to generate interference options. The Press Back action is excluded from the interference generation action space, as the functionality of the back button can vary significantly across different applications. For this task, when the current action is CLICK, a semi-transparent red

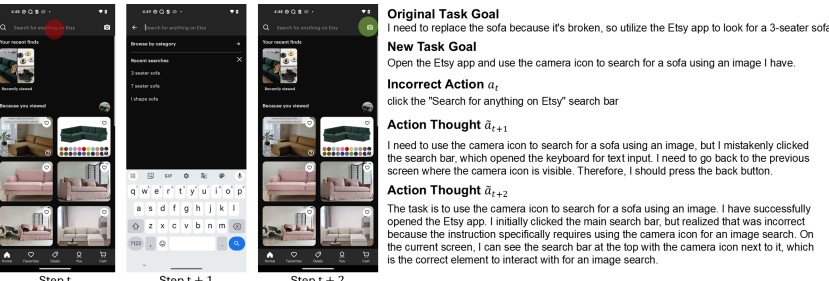

Figure 7: Example of the reflection data generated with the first approach in the offline SFT stage. The click point of the incorrect action is highlighted with a red circle on the first screenshot, and the click point of the action at step $t + 2$ is highlighted with a green circle in the third screenshot.

circle is painted at the click location on the first screenshot, serving as a visual cue. We construct the evaluation and training data from AndroidControl and Odyssey. We have 420 samples in total for the evaluation part and 8642 samples for training. The task template for this task is shown in Table 10.

### F.3 Mistake-informed Reattempt

For this task, we construct training data from the Wave-UI [2], AMEX [11], and OS-ATLAS-Desktop [43] datasets and directly evaluate on the grounding benchmark ScreenSpot [14] and ScreenSpot [43]. To obtain the failed attempts in training data, we first train a GUI-Pretrain model with the three target datasets for this task excluded. Then we conduct inference on these three datasets and select samples with failed predictions. We also use the bounding boxes of other elements annotated on the same image as mistake candidates. To construct mistake-informed training data, for each sample, we randomly choose 1 to 5 failed attempts and provide these mistakes in the prompt. The bounding boxes of the mistakes are also drawn using red rectangles on the image. We have 31836 samples in total for the training of this task. The task template is shown in Table 11.

## G Details of Reflection Data in Offline SFT

We use two approaches to construct reflection data in the offline SFT stage. For the first approach, we first adopt Gemini-2.5-Pro [38] (prompt shown in Table 12) to modify the original goal to make the action incorrect. With the modified goal $\tilde{G}$, we further generate the reflection action at $t + 1$ with the prompt shown in Table 13. If the reflection action $\tilde{a}_{t+1}$ is `Press Back`, we assume $\tilde{I}_{t+2} = I_t$. Then at step $t + 2$, the agent needs to summarize the previous mistake and make an informed new attempt. To obtain such action annotation, we first generate the correct action $\tilde{a}_t$ at step $t$ after the goal is modified to $\tilde{G}$ using the prompt shown in Table 14. The action $\tilde{a}_t$ does not contain the reflection part about the mistake, so we further modify the action thought $\tilde{a}_t^{thought}$ to $\tilde{a}_{t+2}^{thought}$ while keeping the action description and grounded action by adding reflection content using the prompt provided in Table 15.

For the second approach, we first create an ineffective incorrect action using the prompt in Table 16. And then we modify the original $a_t^{thought}$ by adding reflection content about additional inserted ineffective action using the prompt in Table 17.

We build reflection data from the AndroidControl dataset and obtain 17557 samples with the first approach and 15394 samples with the second approach in total. Examples of the reflection data generated via theses two approaches are provided in Fig 7 and Fig 8.

## H Details of Online Iterative Reflection Tuning

The Apps in our online environment with the task statistics and examples are shown in Table 21.

For the MLLM-based verifier, we adopt Gemini-2.0-Flash as the MLLM and provide the complete sequence of screenshots, task goal, task guidance, and action sequence to it for judgment. The prompt for this process is provided in Table 18.

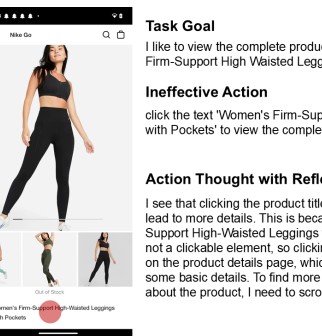

Figure 8: Example of the reflection data generated with the second approach in the offline SFT stage. The click point of the ineffective action is highlighted with a red circle on the first screenshot.

In our online reflection tuning algorithm, we further check the step-wise correctness for those successful trajectories using Gemini-2.0-Flash with the prompt shown in Table 19. For unsuccessful trajectories, we use GPT-4o and the prompt in Table 20 to identify the first error step. The pre-correction and post-reflection annotations are similar to the process used in Sec G with additional task guidance provided for more accurate annotation.

# I  Detailed Experiment Results

We provide detailed evaluation results on ScreenSpot and ScreenSpotv2 in Table 6 and Table 7.

Table 6: Detailed evaluation results on ScreenSpot.

| Model | Mobile | | Desktop | | Web | | Avg |
|---|---|---|---|---|---|---|---|
| | Text | Icon/Widget | Text | Icon/Widget | Text | Icon/Widget | |
| InternVL2.5-8B | 81.32 | 52.40 | 46.39 | 30.00 | 43.48 | 25.24 | 46.47 |
| InternVL3-8B | 91.57 | 75.11 | 76.80 | 52.86 | 77.39 | 55.82 | 71.59 |
| GUI-Pretrain | 92.67 | 74.67 | 91.75 | 72.86 | 90.87 | 78.64 | 83.58 |
| GUI-Pretrain-Ref | 95.24 | 79.48 | 92.78 | 69.29 | 90.43 | 83.50 | 85.12 |

Table 7: Detailed evaluation results on ScreenSpotv2.

| Model | Mobile | | Desktop | | Web | | Avg |
|---|---|---|---|---|---|---|---|
| | Text | Icon/Widget | Text | Icon/Widget | Text | Icon/Widget | |
| InternVL2.5-8B | 80.69 | 57.34 | 45.36 | 29.28 | 35.47 | 27.58 | 45.95 |
| InternVL3-8B | 92.76 | 79.15 | 75.26 | 52.14 | 75.21 | 57.63 | 72.02 |
| GUI-Pretrain | 95.86 | 78.67 | 93.30 | 71.43 | 91.45 | 78.33 | 84.84 |
| GUI-Pretrain-Ref | 97.24 | 83.41 | 94.33 | 70.71 | 92.31 | 83.25 | 86.88 |

Table 8: Prompt for negative purpose annotation

## System Role
You are an expert in understanding GUI operations.

## Information Provided
You will receive the following information:
1. Two screenshots: the first screenshot corresponds to the current step, and the outcome screenshot corresponds to the next step.
2. Current Action: the action executed at the current step
3. Current Action Purpose: The purpose of executing the current action

The current action successfully achieved the current purpose (can be seen from the outcome of this action, which is the outcome screenshot). Your task is to come up with a new purpose, such that it is not satisfied according to the outcome screenshot. Note that your new purpose should be somewhat difficult and confusing, such that someone who is unfamiliar with the GUI operations or functions might think it is satisfied by the current action executed. But make sure your new purpose has no ambiguity so an expert can determine that it is not satisfied based on the screenshots.

Note that the new purpose should be a verb phrase (starting with a verb) describing the overall purpose and it should **not** be some direct low-level action instructions like 'click the xxx'.
Make sure your new purpose is NOT satisfied after the execution of the current step.
!!! Important
Make sure your new purpose can actually be satisfied by doing some **single action** different from the current action on the **first screen**.
You also need to provide a simple explanation about what single action on the current step can satisfy your new purpose and why an expert can tell from the outcome screen that your new purpose is not satisfied.

## Input:
Current Action: {action description}
Current Action Purpose: {action purpose}

## Output Format:
Directly output your results in json format as follows:
{{
"new purpose": "your new purpose",
"explanation": "your explanation"
}}

Table 9: Task template for Action Verification

<image>
<image>
Given an action purpose and two screenshots (the first screenshot corresponds to the step before a certain action while the second one is the outcome screenshot after the execution of the action). You need to judge whether the action purpose has been satisfied by the action executed between these screenshots based on the screenshots content. Directly answer Yes or No.

## Input
action purpose: {action purpose}

## Output:

Table 10: Task template for Action Reversal

<image>
<image>
You are an expert in evaluating the behavior of GUI agents that interact with Android phone interfaces. Your task is to assist in training intelligent agents by identifying the correct **revert operation** to undo a previously executed incorrect operation.

You will be given:
- A **current action** that was performed
- A pair of screenshots:

- **Screen A**: the UI before the action
- **Screen B**: the UI after the action
- A set of **six revert operation action choices (A-F)**
- The agent must choose **the one correct revert operation** that best reverts Screen B back to Screen A.

## Valid Action Space:
- **Open app[app]**: Open the specified app.
- **Click**: Tap on a specific UI element.
- **Long Press**: Long press on a specific UI element.
- **Scroll**: Perform a scroll gesture on the screen.
- **Type[text]**: Input the specified text into a text field.
- **Press Home**: Return to the home screen.
- **Press Back**: Return to the previous screen.
- **Press Enter**: Confirm input using the enter/return key.

## Evaluation Criteria:
- The **correct revert operation** must be the most effective and reasonable way to return the UI from **Screen B** to **Screen A**, based on the change caused by the current action.
- Only **one option** is correct. The remaining five should be plausible but incorrect.
- Evaluate the options based on:
- Whether the action targets the correct UI element
- Whether it reverses the effect of the current action

## Input:

- **Current Action**: {action_desc}

- **Choices**: A-F options {undo_options}

## Output Format:
Directly output the option letter only

Table 11: Task template for Mistake-informed Reattempt

<image>
You are given a screenshot of a mobile phone screen, a question, and some incorrect answers have already been excluded for you.
You need to give a correct answer based on the screenshot. Notice that the correct answer should be different from the incorrect answers.
The question is: {grounding instruction}
The incorrect answers (also annotated using red bbox in the image) are: {incorrect answer}
The correct answer is:

Table 12: Prompt for instruction modification.

## System Role
You are an expert in understanding the operations from a GUI agent. The agent's task is to help human users operate an Android phone by completing specific instructions provided by the user. Your goal is to modify the original instruction such that the action taken by the agent becomes incorrect in the context of your modified instruction.

## Valid Action Space
The actions that the agent may take to complete the task are as follows:
Open app[app]: Open the 'app' APP
Click: Click on the current screen.
Long Press: Long press on the current screen.
Scroll: Scroll on the screen.
Type[text]: Type the 'text' into the input field.
Press Home: Return to the home page.
Press Back: Go back to the previous page.
Press Enter: Click the enter button.
Wait: Wait for the device to respond or load something
Task Complete/Task Impossible: The agent indicates the task is completed or impossible.
## Information Provided
You will receive the following information:
1. Task Instruction: The original task instruction for the GUI agent.
2. Past Actions: The past actions taken by the agent to complete this task before this step. Empty if it is the first step.
3. Future Actions: The future actions taken by the agent to complete this task after this step. Empty if it is the last step.
4. Current Action: The action taken by the agent at the current step.
5. Screenshots: The screenshot at the current step and the screenshot at the next step after the current action is executed. The click point is highlighted using a red dot in the current screenshot and the scroll positions are visualized using a red arrow in the current screenshot.

Your job is to analyze the provided information to provide a modified instruction. With the modified instruction, the current action becomes incorrect, while the past actions are still correct (compatible with the modified instruction).
Incorrect action means that the current action taken by the agent is clearly wrong or unnecessary for completing the modified instruction or deviates from the correct way.

! Important
Your modified instruction should be natural, reasonable, and also realistic. With your modified instruction, the now-incorrect action (**current action**) becomes an easy or natural mistake a user who is unfamiliar with the App, button functions, or certain operations might make. The user will probably realize the mistake when seeing the execution of this action (the second screenshot).

## Useful Guidelines
To make the evaluation process more accurate, please follow these guidelines:
1. Based on the information, judge the feasibility of having a reasonable instruction meeting the mentioned requirements. Ignore the following if not.
2. Provide your modified instruction.
3. Explain why the modified instruction is reasonable and realistic, and does not change the correctness of the past actions.
4. Explain why the **current action** becomes incorrect with your modified action, and explain why it is an easy or natural mistake a user might make.
Note that the red dot and red arrows are just for visualizing the actions; do not mention them in your response.
Note that the future actions are provided to better understand the overall task and context; you do not need to consider them when creating modified instruction.

Note that your modified instruction should clearly be possible for the agent to complete based on the provided information.

## Input:
Task Instruction: {task_instruction}
Past Actions: {past_actions}
Future Actions: {future_actions}
Current Action: {action}

## Output Format:
Directly output your results in json format as follows:
{{
"Task Feasibility": "Yes or No", if no, the following entries should be empty. "Modified Instruction": "Your modified instruction",
"Explanation 1": "Explain why the modified instruction is good and compatible with action history.",
"Explanation 2": "Explain why the current action becomes a natural mistake for a user not familiar with the app and certain operations."
}}

Table 13: Prompt for reflection action annotation $\tilde{a}_{t+1}$ of the first approach in offline SFT.

## System Role
You are an expert in correcting the step-wise operation of a GUI agent. The agent's task is to help human users operate an Android phone by completing specific instructions provided by the user but it performs an incorrect action at a certain step. Your task is to make a reflection about the previous incorrect step and reason about the next correct atomic action after this mistake, provide the action thought, and the action type.

## Definition of Action Thought
The action thought is the rationale behind the actual action taken at a certain step. The action thought should be a compact paragraph consisting of 3-4 sentences. It could include the following aspects if they are helpful and important for the action reasoning process: - observation: a concise description of the current screenshot, focusing on the task related content and progress.
- reflection: a simple analysis of the unexpected situation caused by the incorrect action in the previous step, admit and analyze the mistakes.
- action rationale: a brief reasoning process to integrate the above information and provide a natural thinking process leading to the actual action from the perspective of the operator
The action thought should be natural and logically fluent. The action thought should *not* be clearly separated into the above aspects and does not need to mention the keywords like observation and reflection.

## Information Provided
You will receive the following information:
1. Task Instruction: The overall task instruction for the GUI agent.
2. Past Actions: A list of past actions prior to the previous incorrect step. Empty if it is the first step.
3. Previous Incorrect Action: The incorrect action taken at the previous step.
4. Error Action Analysis: An analysis from an expert explaining why the previous action is incorrect.
5. Screenshots: The first screenshot corresponds to the step before the incorrect action is taken while the second screenshot corresponds to the current step after executing the incorrect action. The click point is highlighted using a red dot in the first screenshot.

Your job is to analyze the provided information and provide the action thought leading the agent should take at the current step after the execution of the previous incorrect action.

## Valid Action Space
The actions that the agent may take to complete the task are as follows:
Click: Click at a certain position on the current screen.
Long Press: Long press at a certain position on the current screen.
Scroll: Scroll on the screen, scroll down/up/left/right, where the direction is the opposite direction of the figure movement.
Type[text]: Type the 'text' into the input field.
Press Home: Return to the home page.
Press Back: Go back to the previous page.
Press Enter: Click the enter button.
Wait: Wait for the device to respond or load something.
Task Complete/Task Impossible: Indicate the task is completed or impossible.

## Useful Guidelines
To make your reasoning and response more accurate, please follow these guidelines:
1. Analyze the provided information
2. Provide your action thought. In the action thought, use first-person description, that is, using 'I' instead of 'the user' or 'the agent'
3. Provide your action type which must be consistent with your action thought.
The correct action type should be one of the following: [Click, Long Press, Scroll, Type, Press Home, Press Back, Press Enter, Wait, Task Complete, Task Impossible]

## Important Notes:
Note that the red dots for click and long press are only for better understanding of the actions; do not mention them in your action thought.
Note that some sub-task might need multiple actions (e.g. typing something needs clicking the text input field and then typing) and you should only give the very first atomic action for the current step.
Note that if the previous step type in some incorrect content, you have to first find ways to clear it before typing the correct one.
Note that when you want to click or press the backspace button, the correct action type is click instead of Press Back (which means going back to the previous page). And in such cases, try to use the word 'click' instead of 'press' in your action thought.

## Input:
Task Instruction: {task_instruction}
Past Actions: {past_actions}
Previous Incorrect Action: {incorrect_action}
Error Action Analysis: {error_analysis}

## Output Format:
Directly output your results in json format as follows:
{{
"Action Thought": "The action thought"
"Action Type": "The action type"
}}

Table 14: Prompt for correct action annotation $\tilde{a}_t$ of the first approach in offline SFT.

## System Role
You are an expert in correcting the step-wise operation of a GUI agent. The agent's task is to help human users operate an Android phone by completing specific instructions provided by the user but it performs an incorrect action at a certain step. Your goal is to reason about the correct action and provide the reasoning process leading to the correct action at this step.

## Definition of Action Thought

The action thought is the rationale behind the actual action taken at a certain step. The action thought should be a compact paragraph consisting of 3-4 sentences. It could include the following aspects if they are helpful and important for the action reasoning process:
- observation: a concise description of the current screenshot, focusing on the task related content
- reflection: a simple analysis of whether the previous step's action is successful and as expected
- progress analysis: a brief summary of the progress towards the overall goal before taking the action at the current step and a plan about the sub-tasks that still need to be done in order to complete the final goal
- action rationale: a brief reasoning process to integrate the above information and provide a natural thinking process leading to the actual action from the perspective of the operator
The action thought should be logically fluent, and it does not necessarily include all the above aspects. The action thought should *not* be clearly separated into the above aspects and does not need to mention the keywords like observation and reflection.

## Information Provided
As an evaluator, you will receive the following information: 1. Task Instruction: The overall task instruction for the GUI agent.
2. Past Actions: A list of past actions prior to this step. Empty if it is the first step.
3. Screenshots: The first screenshot corresponds to the previous step while the second screenshot corresponds to the current step. Only the current screenshot is provided if it is the first step. The click point is highlighted using a red dot in the first screenshot.

Your job is to analyze the provided information and provide the correct action thought leading to the correct action that the agent should take.

## Valid Action Space
The actions that the agent may take to complete the task are as follows:
Click: Click at a certain position on the current screen.
Long Press: Long press at a certain position on the current screen.
Scroll: Scroll on the screen, scroll down/up/left/right, note that the direction is the opposite direction of the figure movement.
Type[text]: Type the 'text' into the input field.
Open App[app]: Open the 'app' App.
Press Home: Return to the home page.
Press Back: Go back to the previous page.
Press Enter: Click the enter button.
Wait: Wait for the device to respond or load something.
Task Complete: Indicate the task is completed
Task Impossible: Indicate the task is impossible.

## Useful Guidelines
To make your reasoning and response more accurate, please follow these guidelines:
1. Based on the provided information, provide the correct action thought. In the action thought, use first-person description, that is, using 'I' instead of 'the user' or 'the agent'
2. Provide the correct action type which must be consistent with your action thought.
The correct action type should be one of the following: [Click, Long Press, Open App, Scroll, Type, Press Home, Press Back, Press Enter, Wait, Task Complete, Task Impossible]

## Important Notes
Note that when you want to click or press the backspace button, the correct action type is click instead of Press Back (which means going back to the previous page). And in such cases, try to use the word 'click' instead of 'press' in your action thought.

## Input:
Task Instruction: {task_instruction}
Past Actions: {past_actions}

## Output Format:
Directly output your results in json format as follows:
{{
"Action Thought": "The correct action thought"
"Action Type": "The correct action type"
}}

Table 15: Prompt for adding reflection to action thought $\tilde{a}_t^{thought}$ of the first approach in offline SFT.

## System Role
You are an expert in correcting the step-wise operation of a GUI agent. The agent's task is to help human users operate an Android phone by completing specific instructions provided by the user. The agent performs an incorrect action at a certain step, and it needs to execute the press back action to go back to the previous normal state. Your task is to add some reflection content about the previous incorrect action to the correct action thought.

## Definition of Action Thought
The action thought is the rationale behind the actual action taken at a certain step. The action thought should be a compact paragraph consisting of 3-4 sentences. It could include the following aspects if they are helpful and important for the action reasoning process:
- observation: a concise description of the current screenshot, focusing on the task related content
- reflection: a simple analysis of whether the previous step's action is successful and as expected
- progress analysis: a brief summary of the progress towards the overall goal before taking the action at the current step and a plan about the sub-tasks that still need to be done in order to complete the final goal
- action rationale: a brief reasoning process to integrate the above information and provide a natural thinking process leading to the actual action from the perspective of the operator
The action thought should be logically fluent, and it does not necessarily include all the above aspects. The action thought should *not* be clearly separated into the above aspects and does not need to mention the keywords like observation and reflection.

## Information Provided
You will receive the following information:
1. Task Instruction: The overall task instruction for the GUI agent.
2. Incorrect Action: The previous incorrect action
3. Error Action Analysis: An analysis from an expert explaining why the previous action is incorrect.
4. Correct Action Thought: The actual correct action thought leading to the correct action
5. Screenshot: The screenshot after performing the press back action to go back to the normal status

Your job is to analyze the provided information and modify the correct action thought by adding some reflection and lessons learned about the incorrect action narrated in the first person as if you have performed the previous incorrect action, realize the mistake, and press back to go to the current status.

## Important Notes:
Your modified action thought should still be consistent with the provided correct action thought.

## Input:
Task Instruction: {task_instruction}
Incorrect Action: {incorrect_action}
Error Action Analysis: {error_analysis}
Correct Action Thought: {correct_action_thought}

## Output Format:
Directly output your results in json format as follows:
{{
"Updated Action Thought": "The updated action thought with reflection added"
}}

Table 16: Prompt for creating ineffective incorrect action of the second approach in offline SFT.

## System Role
You are an expert in understanding the operations from a GUI agent. The agent's task is to help human users operate an Android phone by completing specific instructions provided by the user. Your goal is to come up with an incorrect ineffective Action that is different from the correct action and would not change the current screen.

## Candidate Actions
You may consider the following actions
- click: click on the current screen.
The complete format of this action is "click <element> to <purpose>", where <element> is a noun phrase starting with 'the' indicating the clicked element, and <purpose> is a verb phrase indicating the direct purpose and expected outcome of clicking this UI element.

- long press: long press on the current screen.
The complete format of this action is "long press <element> to <purpose>", where <element> is a noun phrase starting with 'the' indicating the pressed element, and <purpose> is a verb phrase indicating the direct purpose and expected outcome of long pressing this UI element.

- scroll: scroll on the screen.
The complete format of this action is "scroll <direction> to <purpose>", where <direction> is one of the following directions: ['up', 'down', 'left', 'right'] and the direction is the opposite direction of the movement of the finger. <purpose> is a verb phrase indicating the direct purpose and expected outcome of this scrolling action.

- type: type some 'text' into the input field.
The complete format of this action is "type in the <content>", where <content> is the typed text.

## Information Provided
You will receive the following information: 1. Task Instruction: The original task instruction for the GUI agent.
2. Past Actions: The past actions taken by the agent to complete this task before this step. Empty if it is the first step.
3. Future Actions: The future actions taken by the agent to complete this task after this step. Empty if it is the last step.
4. Current Action: The action taken by the agent at the current step.
5. Screenshot: The screenshot at the current step. The click point is highlighted using a red dot in the current screenshot.

Your job is to analyze the provided information and decide whether it is possible to come up with an incorrect ineffective action.
The requirement of the incorrect ineffective action:
1. The action is incorrect; it is clearly wrong or unnecessary for completing the task or deviates from the correct way.
2. It is an easy or natural mistake a user who is unfamiliar with the App, button functions, or certain operations might make. The user will probably realize the mistake when seeing the execution of this action.

3. This action has no effect on the current screen, which means the current screen will remain exactly the same after executing the incorrect action.

Some examples: scroll down while it is already at the bottom, click the entry name instead of the actual text field for entering information, type in something without activating the input field yet.

## Useful Guidelines
To make the evaluation process more accurate, please follow these guidelines:
1. Based on the information, judge the feasibility of having an incorrect ineffective action meeting the mentioned requirements. Ignore the following if not.
2. Provide your incorrect ineffective action if possible.
3. Explain why the incorrect ineffective action is incorrect, and explain why it is an easy or natural mistake a user might make.

Note that you have to strictly follow the complete format for your action.

## Input:
Task Instruction: {task_instruction}
Past Actions: {past_actions}
Future Actions: {future_actions}
Current Action: {action}

## Output Format:
Directly output your results in json format as follows:
{{
"Task Feasibility": "Yes or No", if no, the following entries should be empty.
"Incorrect Ineffective Action": "Your incorrect ineffective action",
"Explanation": "Explain why the action is incorrect and why it is a natural mistake for a user not familiar with the app and certain operations."
}}

Table 17: Prompt for adding reflection about the ineffective incorrect action of the second approach in offline SFT.

## System Role
You are an expert in correcting the step-wise operation of a GUI agent. The agent's task is to help human users operate an Android phone by completing specific instructions provided by the user. The agent performs an incorrect action at a certain step, and it needs to realize the mistake and perform the correct action. Your task is to add some reflection content about the previous incorrect action to the correct action thought.

## Definition of Action Thought
The action thought is the rationale behind the actual action taken at a certain step. The action thought should be a compact paragraph consisting of 3-4 sentences. It could include the following aspects if they are helpful and important for the action reasoning process:
- observation: a concise description of the current screenshot, focusing on the task related content
- reflection: a simple analysis of whether the previous step's action is successful and as expected
- progress analysis: a brief summary of the progress towards the overall goal before taking the action at the current step and a plan about the sub-tasks that still need to be done in order to complete the final goal
- action rationale: a brief reasoning process to integrate the above information and provide a natural thinking process leading to the actual action from the perspective of the operator
The action thought should be logically fluent, and it does not necessarily include all the above aspects. The action thought should *not* be clearly separated into the above aspects and does not need to mention the keywords like observation and reflection.

## Information Provided
You will receive the following information:
1. Task Instruction: The overall task instruction for the GUI agent.
2. Incorrect Action: The previous incorrect action 3. Error Action Analysis: An analysis from an expert explaining why the previous action is incorrect and why it might happen.
4. Correct Action Thought: The actual correct action thought leading to the correct action
5. Screenshot: The screenshot of the current step after executing the incorrect action

Your job is to analyze the provided information and modify the correct action thought by adding some observation and reflection content realizing and acknowledging the incorrect action narrated in the first person as if you have performed the previous incorrect action.

## Important Notes:
The previous incorrect action has no effect on the screenshot; that is, the screenshots before and after the incorrect action are the same.
Your modified action thought should still be consistent with the provided correct action thought.
!!! Note that the correct action thought does not include the previous incorrect action, so the 'previous step' in the correct action thought actually corresponds to the step before the previous incorrect action. So you have to **remove** the part describing the success of the previous step or description about the progress in the original correction action thought! For example, you should remove parts like 'I have successfully xxx' or 'The previous step has successfully xxx'.
!!! In the first sentence of your thought, you should directly mention that the previous step is unsuccessful or incorrect by observation, and then do a reflection explaining why you made that mistake and the fact learned from this failure.

## Input:
Task Instruction: {task_instruction}
Incorrect Action: {incorrect_action}
Error Action Analysis: {error_analysis}
Correct Action Thought: {correct_action_thought}

## Output Format:
Directly output your results in json format as follows:
{{
"Updated Action Thought": "The updated action thought with reflection added"
}}

---

Table 18: Prompt for the MLLM-based verifier.

---

## System Role
You are an expert in evaluating the performance of a GUI operation agent. The agent's task is to help human users operate an Android phone by completing specific instructions provided by the user. Your goal is to evaluate whether the agent successfully completed the task or not.

## Information Provided
As an evaluator, you will receive the following information:
1. Task Instruction: The original task instruction for the GUI agent.
2: Task Guidance: An overall description about how to **correctly** complete this task as a reference.
3: Correct Answer: The ground truth answer for this task. Empty if the answer is not available or the task does not require an answer.
4. Operation History: A list of actions (in the form of images and corresponding actions) taken by the agent to execute the Task Instruction.

5. Agent Answer: The answer provided by the agent for the task. Empty if there is no answer action.
6. Final Status: The final task status is indicated by the agent. The status is either Task Complete or Task Impossible.

Your job is to analyze the provided information to determine whether the agent successfully completed the task, based on the alignment between the Task Instruction, the actions performed (shown in screenshots and operation history), and the expected outcome.

## Effective Action Space
The actions that the agent may take to complete the task are as follows:
Click[[x, y]]: Click the location [x, y] on the current screen, marked with a red circle in the screenshot.
Long Press[[x, y]]: Click the location [x, y] on the current screen, marked with a red circle in the screenshot.
Scroll[[x1, y1, x2, y2]]: Scroll from position [x1, y1] to [x2, y2], as shown by the arrows in the screenshot, to access information that is not currently visible.
Type[text]: Type the 'text' into the input field.
Memorize[text]: Store some 'text' into an intermediate memory for future reference.
Answer[text]: The agent provides the 'text' as the answer.
Press Home: Return to the home page.
Press Back: Go back to the previous page.
Press Enter: Click the enter button.
Wait: Wait for the device to respond or load something
Task Complete/Task Impossible: The agent indicates the task is completed or impossible.

## Task Evaluation Criteria:
SUCCESS: The agent successfully performs all the necessary actions to meet the Task Instruction.
NOT SUCCESS: The agent failed to perform at least one necessary action, or the task could not be completed correctly, based on the screenshots and operation history.

## Useful Guidelines
To make the evaluation process more accurate, please follow these guidelines:
1. Analyze the Task Instruction: You should first analyze what critical milestones have to be completed and what necessary outcomes are expected for this task.
2. Analyze the agent's operations, provide a definitive verdict on whether the task has been successfully completed together your reasoning process.
3. Provide your final answer: 'SUCCESS' or 'NOT SUCCESS'

## Important Notes
When the task instruction asks a certain question or seeks certain information, the agent has to provide the **correct** answer before completing the task, otherwise, the task should be NOT SUCCESS.
When the agent indicates the task is impossible, you should judge whether this task is indeed impossible to complete (e.g. due to network issues). You should output SUCCESS only when both you and the agent indicate the task is impossible.
You should carefully examine the screenshots to double check whether the operations taken by the agents achieve the desired and expected outcome. Any minor input or format issues should be judged as NOT SUCCESS.
Note that the agent might make some wrong attempts during the process, and it should be judged as SUCCESS as long as the agent has successfully completed the task at the end.

## Input:
Task Instruction: {task_instruction}
Task Guidance: {task_guidance}
Correct Answer: {correct_answer}
Agent Answer: {agent_answer}
Final Status: {final_status}

## Output Format:
Directly output your results in json format as follows:
{{
"Task Analysis": "An analysis of the task",
"Reasoning": "Reason about whether the agent has successfully completed the task",
"Final Result": "SUCCESS" or "NOT SUCCESS"
}}

Table 19: Prompt for checking the step-wise correctness of successful trajectories.

## System Role
You are an expert in evaluating the step-wise operation correctness of a GUI agent. The agent's task is to help human users operate an Android phone by completing specific instructions provided by the user. Your goal is to evaluate whether the current action taken by the agent is correct in terms of completing the overall task instruction.

## Information Provided
As an evaluator, you will receive the following information:
1. Task Instruction: The overall task instruction for the GUI agent.
2. Task Guidance: An overall description of how to correctly complete this task as a reference.
3: GT Answer: The ground truth answer for this task. Empty if the answer is not available or the task does not require an answer.
4: Stored Memory: Information the agent has stored in the intermediate memory from previous steps. Empty if none.
5. Task Progress: A summary describing the current progress of the overall task before taking the action at the current step. The progress is empty if it is the first step.
6. Current Action: The action taken by the agent at the current step.
7. Screenshots: The screenshot at the current step and the screenshot at the next step after the current action is executed. The click point is highlighted using a red dot in the current screenshot and the scroll positions are visualized using a red arrow in the current screenshot.

Your job is to analyze the provided information to determine whether the current action taken by the agent is correct for achieving the overall task goal.

## Valid Action Space
The actions that the agent may take to complete the task are as follows: Click: Click on the current screen, the click point is marked with a red circle in the screenshot.
Long Press: Long press on the current screen, the press point is marked with a red circle in the screenshot.
Scroll: Scroll on the screen, the touch point and lift point of the scroll are marked with a red arrow in the screenshot.
Type[text]: Type the 'text' into the input field.
Memorize[text]: Store some 'text' into an intermediate memory for future reference
Answer[text]: The agent provides the 'text' as the answer.
Press Home: Return to the home page.
Press Back: Go back to the previous page.
Press Enter: Click the enter button.
Wait: Wait for the device to respond or load something
Task Complete/Task Impossible: The agent indicates the task is completed or impossible.

## Task Evaluation Criteria:
CORRECT: The current action taken by the agent is reasonable without skipping any necessary steps and makes correct progress towards completing the overall task instruction.

INCORRECT: The current action taken by the agent is clearly wrong or unnecessary for completing the task, or deviates from the correct way. You should consider whether the outcome of the current action (from the second screenshot) is as expected or not when making the judgment.

## Useful Guidelines
To make the evaluation process more accurate, please follow these guidelines:
1. Based on the provided information, analyze the agent's action, provide a definitive verdict on whether the current action is correct, together with your reasoning process.
2. Provide your conclusion: 'CORRECT' or 'INCORRECT'
3. Based on the current progress and the screenshot for the next step, update the task progress summarizing the overall progress after taking this current action. One or two sentences.

Note that some sub-tasks or expected outcomes might need multiple actions (e.g. typing something needs to click the text input field and then type), and you only need to assess the correctness of the current action.
Note that the 'Current Action' represents the purpose of the agent, which might be inconsistent with the actuation of the action due to its unfamiliarity with certain operations or elements when the action is to click/long press/scroll. In such cases, the 'Current Action' itself may seem correct, but the grounded action is wrong. For example, the agent's action (purpose) is to click App A, but it actually clicks App B as it does not know which icon is App A. So you should also make the judgment based on the actual action and its outcome.
!!! Note that an action should be considered correct if it attempts to recover from a previous mistake or redirect the process back toward the final goal, even if earlier steps were incorrect or unnecessary.

## Input:
Task Instruction: {task_instruction}
Task Guidance: {task_guidance}
GT Answer: {gt_answer}
Stored Memory: {memory}
Task Progress: {prev_progress}
Current Action: {action}

## Output Format:
Directly output your results in json format as follows:
{{
"Reasoning": "Reason about whether the current action is correct",
"Conclusion": "CORRECT" or "INCORRECT",
"Updated Progress": "The updated task progress summary after taking this action."
}}

Table 20: Prompt for identifying the first error step of the unsuccessful trajectories.

## System Role
You are an expert in evaluating the step-wise operation correctness of a GUI agent. The agent's task is to help human users operate an Android phone by completing specific instructions provided by the user. Your goal is to evaluate whether the current action taken by the agent is correct in terms of completing the overall task instruction.

## Information Provided

As an evaluator, you will receive the following information:
1. Task Instruction: The overall task instruction for the GUI agent.
2. Task Guidance: An overall description of how to correctly complete this task as a reference.

3: GT Answer: The ground truth answer for this task. Empty if the answer is not available or the task does not require an answer.

4: Stored Memory: Information the agent has stored in the intermediate memory from previous steps. Empty if none.

5. Task Progress: A summary describing the current progress of the overall task before taking the action at the current step. The progress is empty if it is the first step.

6. Current Action: The action taken by the agent at the current step.

7. Screenshots: The screenshot at the current step and the screenshot at the next step after the current action is executed. The click point is highlighted using a red dot in the current screenshot and the scroll positions are visualized using a red arrow in the current screenshot.

Your job is to analyze the provided information to determine whether the current action taken by the agent is correct for achieving the overall task goal.

## Valid Action Space
The actions that the agent may take to complete the task are as follows:

Click: Click on the current screen, the click point is marked with a red circle in the screenshot.

Long Press: Long press on the current screen, the press point is marked with a red circle in the screenshot.

Scroll: Scroll on the screen, the touch point and lift point of the scroll are marked with a red arrow in the screenshot.

Type[text]: Type the 'text' into the input field.

Memorize[text]: Store some 'text' into an intermediate memory for future reference.

Answer[text]: The agent provides the 'text' as the answer.

Press Home: Return to the home page.

Press Back: Go back to the previous page.

Press Enter: Click the enter button.

Wait: Wait for the device to respond or load something.

Task Complete/Task Impossible: The agent indicates the task is completed or impossible.

## Task Evaluation Criteria:
CORRECT: The current action taken by the agent is reasonable without skipping any necessary steps and makes correct progress towards completing the overall task instruction.

INCORRECT: The current action taken by the agent is clearly wrong or unnecessary for completing the task, or deviates from the correct way. You should consider whether the outcome of the current action (from the second screenshot) is as expected or not when making a judgment.

## Useful Guidelines
To make the evaluation process more accurate, please follow these guidelines:

1. Based on the provided information, analyze the agent's action, provide a definitive verdict on whether the current action is correct, together with your reasoning process.

2. Provide your conclusion: 'CORRECT' or 'INCORRECT'.

3. Based on the current progress and the screenshot for the next step, update the task progress summarizing the overall progress after taking this current action. One or two sentences.

Note that some sub-tasks or expected outcomes might need multiple actions (e.g. typing something needs to click the text input field and then type), and you only need to assess the correctness of the current action.

Note that if the task requires question answering (GT Answer is not empty), the agent must provide the answer before marking the task as complete.

## Input:
Task Instruction: {task_instruction}
Task Guidance: {task_guidance}
GT Answer: {gt_answer}

Stored Memory: {memory}
Task Progress: {prev_progress}
Current Action: {action}

## Output Format:
Directly output your results in json format as follows:
{{
"Reasoning": "Reason about whether the current action is correct",
"Conclusion": "CORRECT" or "INCORRECT",
"Updated Progress": "The updated task progress summary after taking this action."
}}

Table 21: Task examples of our online learning environment.

| Apps | Difficulty Level | Number | Example Task |
|---|---|---|---|
| Simple Calendar Pro | Level-1 | 13 | Change the view to {view} in Simple Calendar Pro. |
| Simple Calendar Pro | Level-2 | 7 | Change the location of event {event1} to be the same as event {event2} in Simple Calendar Pro. |
| Contacts | Level-1 | 12 | Delete the contact named {name} in the Contacts App. |
| Contacts | Level-2 | 9 | Find {name} in the Contacts App, change the phone number to {new number} and email to {new email} |
| Dice | Level-1 | 10 | In the Dice App, roll the dice {n times} times with the default setting. |
| Dice | Level-2 | 4 | In the Dice App, set Dice to 1 and Sides to {n sides}, and roll the dice twice with this setting. Record the results and answer the sum of numbers. |
| FitBook | Level-1 | 11 | Open the FitBook App, delete the entry with the highest calories in the diary tab. |
| FitBook | Level-2 | 8 | Open the FitBook App, find and open the {entry} entry in the Diary tab, then change its date to {n days} days before the original date. |
| Fossify Clock | Level-1 | 15 | Add a clock in {timezone} timezone in the Fossify Clock App. |
| Fossify Clock | Level-2 | 9 | Start all timers and then pause them in the Fossify Clock App. |
| Fossify Messages | Level-1 | 8 | Delete the conversation from {number} in the Fossify Messages App. |
| Fossify Messages | Level-2 | 7 | Send the same message {n times} times to {number} in the Fossify Messages App with the following content: {message} |

| | | | |
|---|---|---|---|
| LibreOffice | Level-1 | 10 | In the LibreOffice Viewer App, go to the {folder} folder in the sdk_gphone64_x86_64 storage area. How many docs are there? Answer with the number only. |
| LibreOffice | Level-2 | 4 | Find the numbers logged in {docfile1} and {docfile2} in the sdk_gphone64_x86_64 storage area of the LibreOffice Viewer App. Answer the sum of the numbers. |
| Markor | Level-1 | 13 | Find all markdown notes in Markor. Answer with their names separated with comma. |
| Markor | Level-2 | 6 | Check the file sizes of the note {name1} and {name2} in Markor. Then answer the larger file size in Bytes (number only without units), for example: 500. |
| ProExpense | Level-1 | 11 | What expenses are logged in the Pro Expense App? Answer the names separated by comma. |
| ProExpense | Level-2 | 9 | Delete all expenses in the Pro Expense App that are higher than {amount}. |
| Broccoli | Level-1 | 13 | Add the following recipe in the Broccoli APP: {recipe}. |
| Broccoli | Level-2 | 9 | In the Broccoli APP, check the preparation times of recipes {title1} and {title2}, and answer the longer preparation time. Please directly answer the time in the following format: XX mins/XX hrs. |
| Chrome (WebShopping) | Level-1 | 9 | Go to {website} using Chrome, search for {product}, what is the overall rating of the first search result? Answer with the number only. |
| Chrome (WebShopping) | Level-2 | 9 | Search for the following items {product list} on {website} and add them to my cart. What is the total price of all the items in my cart? |

