# OpenReview forum: "GUI-Reflection: Empowering Multimodal GUI Models with Self-Reflection Behavior"
_NeurIPS.cc/2025/Conference — NeurIPS 2025 poster_

### Official Review · Reviewer_pU3E · 2025-06-24

**Clarity:** 2
**Significance:** 3
**Originality:** 3
**Rating:** 4
**Confidence:** 3

**Summary:**

The paper introduces GUI-Reflection , a framework that enhances end-to-end multimodal GUI models with self-reflection and error correction capabilities. Unlike traditional GUI models that rely on error-free training data and lack recovery mechanisms, GUI-Reflection integrates reflection behaviors through pre-training, offline supervised fine-tuning, and online reflection tuning. The framework includes a GUI-Reflection Task Suite to explicitly train and evaluate action verification, reversal, and mistake-informed reattempts. It also features an automated data pipeline and a distributed mobile environment for iterative online learning. Experimental results show that GUI-Reflection significantly improves the model’s ability to recognize and recover from errors, achieving strong performance on Android-based tasks and benchmarks like AndroidWorld.

**Questions:**

1.	Why does Action Verification in Figure 3 have two Purposes?

2.	In Section 2.3, since MLLMs can be used to filter the coordinates generated by the GUI model, why not directly use MLLMs to generate the coordinates?

3.	I appreciate the authors’ approach of constructing error data in the first manner during the Offline SFT phase. Nevertheless, I would like to ask whether alternative strategies could be considered for generating erroneous actions—particularly those that are more representative of real-world failure cases. Focusing optimization efforts on actions with very low occurrence probabilities may not lead to meaningful improvements in practical scenarios.

4.	Does Table 1b include results for Qwen series models? I suggest the authors add these for comparison.

5.	In Table 2, "Filtered BC" should be expanded as "Filtered Behavior Cloning" upon its first appearance.

**Ethical Concerns:**

["NO or VERY MINOR ethics concerns only"]

**Final Justification:**

The authors have addressed most of my concerns, and I appreciate their clarifications. However, two key issues remain unresolved: (1) The generalization of the method is not convincingly demonstrated. Full results on the Qwen2.5 series are still missing, and the method was only tested on InternVL, limiting its demonstrated applicability. (2) The strategy for generating incorrect examples is unconvincing. Simply replacing correct answers to create negative cases may not generalize to real-world settings. Given these concerns, I maintain my score of 4.

**Limitations:**

Yes

**Quality:**

3

**Strengths And Weaknesses:**

### Strength:
1.	The paper addresses a highly important and timely problem in GUI agents—building reflection capabilities, which is a critical issue needing to be solved.

2.	The proposed solution by the authors is reasonable and demonstrates a progressive enhancement of the model's reflective abilities.

3.	The experimental results on the dataset validate that the proposed method effectively improves the reflection capabilities of GUI agents.

### Weaknesses：

1.	Many details are missing, and no code is provided, even though the authors include supplementary material in the appendix (e.g., specifics about training loss functions).

2.	The experimental baselines are somewhat limited. I recommend the authors include experiments based on the Qwen series models for more comprehensive evaluation.

---

> ### Author Rebuttal · Authors · 2025-07-31
>
> We thank the reviewer for the thoughtful and encouraging feedback. We appreciate the recognition of our work’s importance and are grateful for the detailed suggestions, which we address below point by point.
>
> ---
>
> > **Q1: Many details are missing (e.g., specifics about training loss functions), and no code is provided.**
>
> **A1:**
> Thanks for your suggestion. We use the standard cross-entropy loss as the loss function for training the MLLM, and we will add more information about training and model details in our next revision. We make sure the code, model, and data are open-sourced and publicly available.
>
> ---
>
> > **Q2: Experimental baselines are somewhat limited. Suggest evaluating the Qwen series models.**
>
> **A2:**
> We agree that broader comparisons help assess generalization across architectures. We additionally conduct the GUI Pre-training with our reflection-related data on Qwen-2.5-VL-7B, and the results are shown below. We can see that, similar to the InternVL-8B baseline, after GUI Pre-training with reflection data, the model's reflection-oriented abilities are effectively improved. We will complete the full training pipeline in our framework (including the offline SFT and online training) with Qwen-2.5-VL-7B and add the corresponding results to our paper.
>
> | Model           | Action Verification | Action Reversal |
> |----------------|------------|--------------|
> | Qwen2.5-VL-7B   | 76.36      | 76.90        |
> | + GUI Ref Pre-training | 90.87      | 89.28        |
>
>
> | Model           | ScreenSpot | ScreenSpotv2 |
> |----------------|------------|--------------|
> | Qwen2.5-VL-7B   | 74.94      | 77.50        |
> | - 2nd attempt  | 79.63 | 82.39 |
> | - 3rd attempt  | 80.44 | 83.22 |
> | - pass@3       | 80.61 | 83.37 |
> | Qwen2.5-VL-7B + GUI Ref Pre-training   | 84.95      | 86.76        |
> | - 2nd attempt  | 91.67 | 92.35 |
> | **- 3rd attempt**  | **93.79** | **94.19**  |
> | - pass@3       | 89.22 | 90.30 |
>
> ---
>
> > **Q3: Why does Action Verification in Figure 3 have two Purposes?**
>
> **A3:**
> As described in Section 2.2 and shown in Figure 3, Purpose 1 corresponds to the actual action goal (positive), while Purpose 2 is a plausible but incorrect goal (negative). In this task, the model is only provided with **one of these two purposes** and needs to determine whether the purpose has been achieved. We will add clarifications about this to avoid misunderstanding.
>
> ---
>
> > **Q4: Why not directly use MLLMs to generate the coordinates?**
>
> **A4:**
> We use MLLMs to filter the coordinates generated by the GUI model by **visualizing the predicted coordinates with some red dots**. However, the MLLMs (e.g., Gemini or GPT-4o) can not directly generate or precisely understand the numerical coordinates, so we use our GUI agent to output the coordinates based on the MLLM-generated action thought as described in Section 2.3.
>
> ---
>
> > **Q5: Could alternative strategies be used to generate more representative failure cases during error data construction?**
>
> **A5:**
> Thank you for this thoughtful suggestion. We believe the core concern lies in the representativeness of error cases, particularly regarding (1) real-world failure scenarios and (2) low-probability actions.
>
> First, collecting real-world failure for GUI cases is inherently challenging and costly, as it requires a lot of human supervision to avoid undesirable consequences or safety risks that erroneous actions on real apps could result in. Thus, we adopt a safer and more efficient strategy to utilize existing offline dataset to generate synthetic but plausible errors.
> Indeed, one of the main challenges in constructing error cases from existing offline datasets is that they only contain successful trajectories, so we **lack the post-state screenshots that would result from actual erroneous actions**. This makes realistic error simulation non-trivial. To address this, we prompt the MLLM not only to modify the original goal but also to design the error cases to resemble natural and plausible human mistakes — such as misclicking visually similar elements or misinterpreting UI semantics. While these are not exhaustively representative, they capture common failure patterns and provide useful supervision for learning reflection behaviors.
>
> Second, we clarify that our goal in the Offline SFT stage is **not to fix model's common mistakes**, but only to proactively activate the model's basic ability to anticipate and detect its own errors. The main goal of this stage is to endow the model with the behaviour to detect and correct errors, and these capabilities are then further refined and expanded in the online reflection tuning stage, where real interaction feedback allows more complex behaviors to emerge.
>
> Looking ahead, we plan to explore the use of **generative models or learned world models** to simulate a wider range of realistic erroneous actions and resulting states, enabling richer error distributions and more robust error-recovery learning.
>
> ---
>
> > **Q6: Add results for Qwen series models in Table 1b.**
>
> **A6:**
> Thanks for your suggestion. We have added the corresponding results. Please see the table in A2.
>
>
> ---
>
> > **Q7: Table 2 should expand “Filtered BC” as “Filtered Behavior Cloning” at first appearance.**
>
> **A7:**
> Thank you for pointing this out — we will revise the text to expand “Filtered BC” as “Filtered Behavior Cloning” upon its first appearance for clarity.

---

> > ### Author Response · Authors · 2025-08-03
> > **Gentle Reminder for Feedback on GUI-Reflection Rebuttal and Discussion**
> >
> > Dear Reviewer pU3E,
> >
> > Thank you very much for your thoughtful review and for recognizing the significance of enhancing reflection capabilities in GUI agents. We would like to briefly highlight the key clarifications and updates we have provided in our rebuttal:
> >
> > - Missing Details and Code: We clarified that our models are trained using standard cross-entropy loss. We will add more training and model specifics in the revised version and ensure that all code, models, and data are publicly released.
> >
> > - Evaluation with Qwen Series Models: As you suggested, we extended our experiments to Qwen2.5-VL-7B and observed consistent gains in reflection-oriented abilities from our GUI-Reflection pre-training. These results are now reported in A2, and we will complete the full training pipeline with Qwen and include it in the next version.
> >
> > - Clarification on Action Verification in Figure 3: We’ve explained in A3 that the task involves two purposes (a positive and a negative one), and the model must verify whether the given purpose has been achieved. We’ll clarify this in the revised text to avoid confusion.
> >
> > - Why MLLMs Do Not Directly Output Coordinates: In A4, we noted that MLLMs are limited in generating precise pixel-level coordinates. Instead, we leverage MLLM-generated thoughts and let our GUI agent determine the coordinates accordingly.
> >
> > - On Generating Realistic Failure Cases: As addressed in A5, we recognize the challenge of collecting real-world erroneous cases and thus adopt a strategy that simulates plausible human-like mistakes (e.g., misclicks, semantic misinterpretation). We also emphasize that the offline SFT stage aims to activate reflection capabilities rather than correct common model errors. Future directions include using learned world models to simulate richer, more representative errors.
> >
> > We sincerely appreciate your suggestions, which have helped us strengthen both our method and presentation. If you have any follow-up questions, we’d be happy to respond.
> >
> > Warm regards,
> > Authors

---

> > ### Comment · Reviewer_pU3E · 2025-08-05
> > **Response to Reubuttal**
> >
> > Thanks for the rebuttal. Most of my concerns have been addressed. I will keep my score unchanged.

---

> > > ### Author Response · Authors · 2025-08-06
> > >
> > > Thank you for the follow-up. We're glad most concerns have been resolved and appreciate the time and effort you put into the review.

---

### Official Review · Reviewer_phbt · 2025-07-03

**Clarity:** 2
**Significance:** 2
**Originality:** 2
**Rating:** 4
**Confidence:** 3

**Summary:**

Noting that current UI control models lack strong abilities to recognize and correct mistakes in executing instructions. To this end, the authors propose 5 new methods for generating training data, spread across 3 proposed training phases:

* A "GUI pre-training" phase along with methods for generating "action verification", "action reversal", and "mistake-informed reattempt" SFT data.
* A "GUI offline SFT" phase, with methods for augmenting existing trajectory data to construct reflection and error correction scenarios.
* A "GUI online learning" phase, an interative process for collecting additional data for error correction via environment interaction.

The authors show that this recipe enable constructing a competitive 8B model on the AndroidWorld datasets. Some qualitative anecdotes of error correction behavior are observed.

**Questions:**

See weaknesses above.

**Ethical Concerns:**

["NO or VERY MINOR ethics concerns only"]

**Final Justification:**

I am a bit split on acceptance. I had significant issues with the presentation and lack of key ablations in the original draft. However, the authors in rebuttal added some of these key ablations and pledged to revise some of the presentation issues. I think the paper could be improved above the acceptance threshold with these changes, but it is difficult to re-evaluate the entire paper in this new context.

I am raising my score to 4, but also think the paper could potentially benefit from re-submission and new round of reviews with the ablations and presentation changes discussed during rebuttal.

**Limitations:**

Yes

**Quality:**

2

**Strengths And Weaknesses:**

__Strengths__

* The limited ability of many current UI control models to recognize and recover from incorrect actions is an important problem to study.
* The authors' proposed methods for generating additional training data focused on this capabilitiy may provide useful, reusable approaches for improving performance.
* The authors demonstrate that their proposed recipe enables competitive performance on the AndroidWorld benchmark.

__Weaknesses__

While the paper addresses an important problem, and some of the proposed methods may in fact be quite useful for addressing it, I did have some issues with the current manuscript and set of results that I think would benefit from revisions and resubmission. The proposed training pipeline is quite complex, and the paper proposes many new methods for data augmentation. However, the experiments don't seem to sufficiently justify the proposed complexity and diversity of methods relative to prior work. The paper reads as a collection of ad-hoc methods, without clear take-aways for which aspects of the overall recipe are most useful for future work to adopt, and why.

More specifically:

* The pipeline involves 3 distinct training phases in addition to the base model's pre-training. However, there is only one end-to-end result in the main paper on a standard end-to-end benchmark: the AndroidWorld result in Table 3. To justify the complexity of the proposed approach, it would have been useful to show clearer ablations for how the various components affect end-task performance. The single competitive-but-not-SOTA result on AndroidWorld is not particularly useful for understanding this. How does each of the proposed datasets and training phases affect end-task performance? Can the recipe be simplified or the number of training phases reduced? These are questions that would have been helpful to address in the paper.
* In general, the paper would perhaps read better if it was more focused on a smaller set of methods and contributions. If there are some aspects of the overall recipe that do not significantly impact end-task performance, perhaps these could be relegated to the appendix.
* Similarly, the number of different proposed training sets and training phases also made the exposition of the overall recipe, and the presentation of the various intermediate results in Tables 1 and 2, a bit difficult to follow at times.
* Key questions related to the effectiveness of the proposed method for "Iterative Online Reflection Tuning" relative to prior work don't appear to be addressed. For example, how important is it that this method uses on-policy sampling from the student model vs. sampling trajectories from a more powerful MLLM (more akin to a standard distillation procedure)? How does the proposed method compare to more standard rejection sampling? Ablations comparing end-to-end task performance would be useful to answer these questions. Is simply scaling up this training stage sufficient towards improving error correction abilities? Focusing on answering these questions alone would have been an interesting and useful paper.

Minor nits:

* The details in section 2.3 of how existing models were used for data annotation were a bit sparse, perhaps there is an Appendix section to reference?
* "pre-training stage" -> This is seemingly an unconventional and confusing name for a stage that occurs after the base model has been pre-trained (InternVL2.5-8B)
* The terminology "agent-based" vs. "end-to-end" seemed a bit strange, given that all approaches for UI control are "agents" in some sense. Would "prompted" vs. "task-specific tuning" be accurate?
* Line 54 in Introduction - "further RL training cannot effectively activate or enhance such reflection abilities" - citation needed? This claim seems questionable in general.
* While the examples in Figure 6 are useful, it would be useful, if possible,  to quantify the improvement in error-correction behavior over a relevant baseline on AndroidWorld.
* (Not a weakness, just a note) Looking at the AndroidWorld leaderboard, there do appear to be 8B models that outperform the proposed model (e.g. ScaleTrack, V-Droid), but these appear to be released within the last couple months, and therefore should be considered concurrent work.

---

> ### Author Rebuttal · Authors · 2025-07-31
>
> We thank the reviewer for the detailed and thoughtful feedback. We appreciate the recognition of our goal to improve error correction in GUI agents and the suggestions to improve the clarity and focus of the paper. We address each point raised below.
>
> ---
>
> > **Q1: Only one end-to-end benchmark result is shown; lack of ablations makes it hard to assess the impact of individual stages or components.**
>
> **A1:**
> We appreciate this insightful suggestion. We agree that providing clear ablations is important to demonstrate the necessity and effectiveness of each proposed training stage. In addition to AndroidWorld, we emphasize that our proposed mobile GUI environment (introduced in Section 2.5.1) also serves as a practical and diverse end-to-end evaluation suite. In Section 3.3 and Table 2, we present an end-to-end evaluation on our environment that verifies the effectiveness of adding reflection-related components at different stages.
>
> To further address the reviewer's suggestion, we have also conducted additional ablations on AndroidWorld, where we systematically compare models with and without reflection-related designs in each stage of our overall pipeline. As shown in the table below, for the three stages of our training pipeline (GUI pre-training, GUI offline SFT, and GUI online training), removing reflection-related design in any stage harms final end-to-end performance, and the reflection-related design in the offline SFT and online stages has larger importance.
>
> |Exp|SR on AndroidWorld (%)|
> |------------------|-------|
> |No Reflection in GUI Pre-training|31.9|
> |No Reflection in offline SFT|26.1|
> |No Reflection in online training|25.9|
> |No Reflection in any stage|20.7|
> |**GUI-Reflection**|**34.5**|
>
> These results collectively validate the **effectiveness of our GUI-Reflection framework as an integrated whole**, and reaffirm our core motivation that reflection and error-correction capabilities should be explicitly injected across all stages of the GUI training pipeline.
>
> ---
>
> > **Q2: Paper may benefit from focusing on fewer methods; some parts could be relegated to the appendix; intermediate results in Tables 1 and 2 make the paper hard to follow.**
>
> **A2:**
> Thank you for the suggestion. We will improve the organization and presentation of our paper to enhance clarity. We will move less important content into the appendix and streamline the exposition of our pipeline to highlight the core insights and contributions.
> Our framework is a **progressive learning process** instead of a set of isolated data augmentations. Specifically: 1) In the GUI pre-training stage, the model learns **basic reflection-oriented atomic skills**. 2) In the GUI offline SFT stage, the model learns to exhibit **reflection behaviors**. 3) In the online training stage, the model learns to **actively reflect and self-correct through real-time interaction**, improving generalization and robustness under open-ended execution.
>
> For the experiments on the GUI-Reflection Task Suite in Tables 1 & 2, our goal is to show that regular GUI pre-training significantly weakens reflection-related capabilities. We introduce this task suite to both evaluate and recover such abilities at the pre-training stage. Beyond this, the GUI-Reflection Task Suite serves as a **complementary benchmark for general-purpose GUI models**. While most existing GUI benchmarks focus on grounding and UI understanding, our tasks emphasize higher-level reasoning skills such as verifying action correctness and reversing errors. These reflection-oriented abilities are crucial for building self-correctable GUI agents and are also useful in broader scenarios—for instance, as a verifier or reward model to supervise intermediate steps.
>
> ---
>
> > **Q3: Effectiveness of the proposed method for "Iterative Online Reflection Tuning"**
>
> **A3:**
> Thanks for your constructive suggestions. We additionally include two baselines: 1) standard rejection sampling, and 2) sampling trajectories from a more powerful MLLM (distillation), and compare them with our proposed method in the online stage and evaluate the end-to-end performance on AndroidWorld. We evaluate these alternatives under a controlled setting where the total number of online training samples is kept similar across methods. For the distillation baseline, we use **Seed1.5-VL** (achieve 62.1 on AndroidWorld) to generate trajectories. The final results are shown in the table below.
> |Baseline| SR on AndroidWorld (%)|
> |------------------|-------|
> |Standard Rejection Sampling|25.9|
> |Distillation|22.4|
> |Ours|**34.5**|
>
> We observe that these baselines exhibit the following limitations:
> - **Standard rejection sampling** is inefficient for complex tasks involving one or several key steps in which the model might get stuck. The model may repeatedly fail to discover successful actions on its own. In contrast, our method enables the model to receive direct feedback on critical error steps, including pre-error correction and post-error reflection guidance from an MLLM, resulting in more efficient and focused improvement.
> - For **distillation**, we observe several challenges:
>   1. If using prompted agentic methods to produce high-quality trajectories, the cost is prohibitively high, making it infeasible to scale.
>   2. Even with strong end-to-end models like Seed1.5-VL, we found that their trajectories **lack explicit reflection behavior**. Through manual inspection and keyword analysis (e.g., "mistake", "incorrect", "unsuccessful"), we observe almost no signs of self-correction or reflective reasoning, even though Seed1.5-VL is trained on data sources similar to UI-TARS, which include some human-annotated annotations.
>   3. The student model design has to be aligned with the teacher model. For example, in our original design, the agent model predicts the thought, action description, and action, but the Seed1.5-VL only predicts the thought and final action. Therefore, we have to redesign the output format and action space of our model to be aligned with the teacher model. However, there still exist some issues where the style of the reasoning thought might be inconsistent with the one in the offline stage.
>   4. As a result, the effect of distillation in this case is similar to **simply adding more offline SFT data**, without the benefits of online reflective feedback. This leads to subpar improvements compared to our iterative reflection tuning.
>
> Overall, these comparisons further support the necessity of our design, where targeted correction and reflection behaviors are injected into the learning loop in a way that neither rejection sampling nor distillation can easily replicate. We will continue to expand experiments in this crucial stage to provide more comprehensive insights in the revised version.
>
> ---
>
> ### Minor Comments
>
> - **Q4:** Section 2.3 details are sparse
>   **A4:** Thanks for pointing this out. We will improve this section and add a pointer to the appendix where the detailed annotation pipelines are described.
>
> - **Q5:** "pre-training stage" phrasing is confusing
>   **A5:** Thank you for the feedback. In the GUI agent literature, the term "pre-training" typically refers to the GUI-specific pre-training stage applied on top of a base MLLM, and this usage is common in prior work. To avoid confusion, we will explicitly revise the phrasing to "GUI pre-training" throughout the paper to make this distinction clearer to readers.
>
> - **Q6:** Terminology "agent-based" vs. "end-to-end" seems unclear
>   **A6:** Thank you for the suggestion — we agree that the terminology can be improved. In the revised version, we will adopt the clearer phrasing of "prompted agents" versus "task-specifically tuned agents", which more accurately reflects the distinction we intend to draw.
>
> - **Q7:** Line 54 claim about RL not helping reflection lacks citation
>   **A7:** Our statement was specifically intended to describe the behavior we observed **within the end-to-end GUI training paradigm**. In our experiments, we found that GUI models trained with standard offline SFT on successful trajectories always fail to exhibit reflection or self-correction behaviors (please also see quantitative results in **A8**); therefore, we state that subsequent RL training alone does not reliably recover these capabilities, as such behaviors can not be obtained by exploration (sampling at a high temperature). This empirical observation was a key motivation for our GUI-Reflection framework.
> We will revise the sentence to make this scope and statement clearer, and we will supplement it with supporting results or qualitative examples to avoid misunderstanding and overgeneralization.
>
> - **Q8:** Quantify the improvement in error-correction behavior over a relevant baseline on AndroidWorld.
>   **A8:** Thank you for the suggestion. To provide quantification results, we analyze the **occurrence of reflection-related behaviors** in the model's action thoughts on AndroidWorld. Specifically, we measure how often the model generates thoughts containing keywords indicative of self-reflection or error awareness (e.g., "mistakenly", "unsuccessful", "incorrect", etc.) While this does not directly assess the correctness of reflection, it serves as a proxy signal for the emergence of such behaviors. The ratio of the occurrence of reflection-related behaviors in our GUI-Reflection model is **14.2%**, while the baseline without reflection-related content in all stages is **0%**.
>
> - **Q9:** Clarify whether some 8B baselines released recently outperform your model
>   **A9:** Thank you — we will include the discussion of these related concurrent methods in the next version.

---

> > ### Author Response · Authors · 2025-08-03
> > **Gentle Reminder for Feedback on GUI-Reflection Rebuttal and Discussion**
> >
> > Dear Reviewer phbt,
> >
> > Thank you once again for your detailed and constructive review of our paper. We deeply appreciate your insights and would like to briefly summarize the clarifications and updates we have provided in our rebuttal:
> >
> > - Effectiveness of Each Stage: In response to your concern about pipeline complexity, we conducted detailed ablation studies (see Table in A1) across all three proposed stages on AndroidWorld. The results clearly show that removing reflection-related components at any stage reduces performance, validating the necessity of each component.
> >
> > - Clarified Purpose of Task Suite: Tables 1 & 2 aim to demonstrate how reflection abilities decline after standard GUI pre-training and how our task suite helps recover them. These tasks also serve as a complementary benchmark for high-level reasoning beyond grounding/UI understanding.
> >
> > - Comparison with Distillation and Rejection Sampling: In A3, we added new experiments comparing our iterative online tuning with standard rejection sampling and large-model distillation. Our method achieves a significantly higher success rate (34.5% vs. 25.9%/22.4%), supported by an in-depth analysis of why other methods fall short in encouraging true reflection.
> >
> > - Terminology and Writing Improvements: We acknowledged and will revise several phrasing issues (e.g., “GUI pre-training,” “agent-based”), and refine claims with additional empirical support as per your suggestions.
> >
> > We truly value your time and feedback. If you have any further questions or would like us to elaborate on any specific point, we’d be happy to follow up. Your early thoughts before the final decision would be immensely appreciated.
> >
> > Warm regards,
> > Authors

---

> > > ### Comment · Reviewer_phbt · 2025-08-05
> > >
> > > Thank you for your response. The new ablations address some key questions I had, and proposed changes to the presentation could significantly improve the paper. Therefore I am raising my score to 4 with cautious optimism about the revised version.
> > >
> > > (However, I still think the paper could potentially benefit from resubmission with these changes implemented to fully evaluate the paper in this new context.)

---

> > > > ### Author Response · Authors · 2025-08-06
> > > >
> > > > Thank you for the thoughtful feedback and for raising your score. We will ensure the revised version fully incorporates the proposed changes and improved clarity.

---

### Official Review · Reviewer_DG2g · 2025-07-05

**Clarity:** 3
**Significance:** 2
**Originality:** 3
**Rating:** 4
**Confidence:** 4

**Summary:**

The paper introduces GUI-Reflection, a framework that adds explicit self-reflection and error-correction behaviours to end-to-end multimodal GUI agents. During GUI-specific pre-training, the authors inject three synthetic tasks—Action Verification, Action Reversal and Mistake-Informed Re-attempt—built from existing successful trajectories. In offline SFT, they automatically augment clean demonstrations with reflection episodes that contain (i) the original wrong action, (ii) a natural-language action-thought explaining the error, and (iii) pre- and post-correction actions. An online reflection-tuning loop runs in a mobile-device sandbox: after every rollout, mistakes are detected, labelled and verified by an MLLM-based checker, then fed back for further fine-tuning. On the authors’ suite of 18 mobile tasks, success on level-2 tasks jumps from 14.6 % → 34.7 % after three reflection-tuning iterations, and the final 8 B-parameter model reaches 34.5 % success on the public AndroidWorld benchmark—outperforming all prior end-to-end 8 B models and rivaling much larger systems (e.g. UI-TARS-72 B).

**Questions:**

What fraction of synthetic corrections are rejected by the programmatic / MLLM checker, and what is the observed false-accept rate? A small human audit (e.g. 100~200 samples) would ground confidence in the pipeline.

**Ethical Concerns:**

["NO or VERY MINOR ethics concerns only"]

**Final Justification:**

Thank the author for the responses, I have raised my rating to 4 since authors address many of my concerns.
That being said, I am still concerned about several points that need further experiments:
1. method robustness to model scale and base performance.
2. benchmark varieties. As of today, MLLM-as-a-judge can not be fully trusted on complex real-world task, and the nearly 100% judge success rate on the relatively simple AndroidWorld task largely contributes to the effectiveness of the model. I doubt if MLLM-as-a-judge can still work in this method for more difficult tasks.

Addressing about concerns by empirical evidences could greatly strengthen the paper.

**Limitations:**

See weaknesses

**Quality:**

3

**Strengths And Weaknesses:**

Pros:

• First GUI agent to demonstrate emergent self-reflection without human feedback, a capability likely needed for reliable automation.

• End-to-end data pipeline—no human labour after initial success logs.

• Competitive results on a public benchmark.

Cons:

• Only 8b model is tested, no evidence of the method effectiveness on various scales, especially larger ones where true performance bottleneck emerges.

• Reflection gains measured on synthetic tasks; real app generalisation unclear.

• Scope limited to Android mobile; no evidence on web/desktop/iOS GUIs.

• Eval relies on MLLM judge, and verification precision/recall not quantified, leading to risk of error accumulation.

---

> ### Author Rebuttal · Authors · 2025-07-31
>
> We thank the reviewer for the detailed feedback and constructive suggestions. Below, we address each point raised.
>
> ---
>
> > **Q1: Only 8B model is tested; no scaling analysis is provided.**
>
> **A1:**
> We have additionally trained InternVL3-78B with our proposed GUI-Reflection framework. The model achieves a success rate of 43.1% on the AndroidWorld benchmark, which is comparable to UI-TARS-72B (46.6%), despite UI-TARS being trained with a significantly larger and more diverse set of trajectories. We also train an InternVL3-78B baseline without the reflection-related designs and it only achieves 31.5%. This result highlights the effectiveness and scalability of our reflection-based training pipeline. We acknowledge that the overall performance of GUI agents is jointly influenced by different factors like **capacity of the base model** and the **scale and diversity of the training data**. Investigating these factors in isolation and exploring more systematic scaling trends will be an important direction for future work.
>
> ---
>
> > **Q2: Reflection gains measured on synthetic tasks**
>
> **A2:**
> We would like to clarify that the apps used in both our environment and the AndroidWorld benchmark are **real, commonly used Android apps**, and the tasks are carefully designed to mirror practical, real-life usage scenarios such as calendar scheduling, file management, and document editing. This ensures that our evaluations reflect the model’s effectiveness in realistic contexts, rather than synthetic toy settings. Additionally, for research reproducibility, safety, and better controllability, we choose open-source apps where we have full access to low-level APIs, system states, and internal databases. This enables accurate evaluation and flexible task designs, which are difficult to achieve with closed-source or proprietary applications.
> We fully agree that evaluating GUI agents in more diverse and naturalistic settings is important. Extending our environment to include a broader set of real-world apps is an active direction and will be prioritized in future work.
>
> ---
>
> > **Q3: Scope limited to Android mobile; no web/desktop/iOS evidence**
>
> **A3:**
> We agree that extending to other platforms is valuable. Our proposed GUI-Reflection framework is **platform-agnostic by design**, and the core methodology — including the reflection task suite, automatic correction data generation, and iterative online tuning — is applicable to a broad range of GUI environments, including desktop and web interfaces. Due to resource and engineering constraints, our current experiments focus on the Android mobile platform, which offers a practical starting point with a rich set of apps and well-supported emulation tools. Nevertheless, we view this as a first step toward a more general and cross-platform framework, and we plan to extend our method to other platforms such as desktop and web GUI environments in future work.
>
> ---
>
> > **Q4: Eval relies on MLLM judge; fraction of synthetic corrections rejected & observed false-accept rate**
>
> **A4:**
> Thank you for your valuable suggestions. In our online reflection tuning process, the MLLM-based judgment and annotation are primarily used in the following parts:
> **1)** ***An MLLM-based verifier judges whether a task is completed by the agent.***
> We have 215 task templates in our proposed environment, and most of them (202 out of 215) are evaluated by the **programmatic verifier**, ensuring correctness in judgment. Only 13 tasks require evaluation by the **MLLM-based verifier** (Gemini-2.0 Flash) because they involve intermediate results, which are not feasible for a programmatic verifier. For each task template, we provide detailed task guidance (including task descriptions and instructions) to the MLLM verifier to ensure accurate judgment. We randomly sampled 100 MLLM judgment samples and manually checked their correctness, achieving **100% accuracy** in this part.
> **2)** ***An MLLM-based verifier checks the step-wise correctness of the successful trajectories.***
> For trajectories deemed successful by the programmatic or MLLM-based verifier, we use another MLLM-based verifier (Gemini-2.0 Flash) to assess the correctness of each step. Only steps verified as correct are retained for training. We randomly sampled 200 step-wise samples and manually checked their correctness. The results are summarized below (Note: we denote **correct** steps as positive here):
> | Metric           | Value |
> |------------------|-------|
> | True Positive (TP)  | 187   |
> | True Negative (TN)  | 6   |
> | False Positive (FP) | 1   |
> | False Negative (FN) | 6   |
> | Precision            | 99.5%  |
> | Recall               | 96.9%  |
>
> The verifier achieves **near-perfect Precision (99.5%)**, ensuring that the quality of the step-wise samples for training is maintained. Note that we can be **overly strict** here (a lower Recall is acceptable) since the number of discarded samples remains relatively small.
>
> **3)** ***An MLLM-based checker identifies the first incorrect step of unsuccessful trajectories***
> For unsuccessful trajectories, we use an MLLM-based checker (GPT-4o) to identify the first incorrect step. We randomly sampled 200 step-wise samples and manually checked the correctness of the judgment from the MLLM-based checker. The results are summarized below (Note: we denote **correct** steps as positive here), and we find that the MLLM-based checker ensures high quality in this process:
> | Metric           | Value |
> |------------------|-------|
> | True Positive (TP)  | 162   |
> | True Negative (TN)  | 31   |
> | False Positive (FP) | 3   |
> | False Negative (FN) | 4   |
> | Precision            | 98.2%  |
> | Recall               | 97.6%  |
>
> **4)** ***An MLLM annotates the pre- and post-error corrections for incorrect steps.***
> For incorrect steps in unsuccessful trajectories, we use an MLLM (Gemini-2.5-Pro-Exp) to annotate both pre-error and post-error corrections. We also provide carefully designed task guidance and prompts to help the MLLM generate high-quality annotations. We randomly sampled 200 MLLM-annotated samples and manually checked their correctness. The accuracy of the MLLM annotation is **97.5%**. Note that we **do not use an additional MLLM-based checker** to double-check the annotation, as the MLLM annotator already achieves high accuracy in this task.

---

> > ### Author Response · Authors · 2025-08-03
> > **Gentle Reminder for Feedback on GUI-Reflection Rebuttal and Discussion**
> >
> > Dear Reviewer DG2g,
> >
> > Thank you again for your thoughtful and detailed review of our paper. We sincerely appreciate your valuable feedback. In our rebuttal, we have addressed all of your concerns, including model scaling, real-world generalization, platform scope, and MLLM verification reliability. For your convenience, we briefly highlight our clarifications below:
> > - Scalability: GUI-Reflection significantly improves performance even on large models. Our InternVL3-78B model achieves 43.1% on AndroidWorld, outperforming its non-reflection variant (31.5%) and rivaling UI-TARS-72B (46.6%).
> > - Real-World Generalization: Our tasks are based on real Android apps designed to reflect practical usage scenarios. We also choose open-source apps for safe, reproducible, and controlled evaluation.
> > - Platform Scope: Our method is platform-agnostic and designed to generalize to web/desktop GUIs. Current experiments focus on Android due to practicality and emulator support, but cross-platform extensions are underway.
> > - MLLM Verifier Accuracy: We conducted extensive audits across all MLLM-involved processes. The results validate the reliability of our pipeline.
> >
> > We welcome any follow-up thoughts or questions you may have. Your early feedback before the decision phase would be extremely valuable to us.
> >
> > Best regards,
> > Authors

---

> > > ### Comment · Reviewer_DG2g · 2025-08-05
> > > **Reviewer reponse**
> > >
> > > Thanks the author for the calrifications, they address many of my concerns.
> > > I am still concerned about the experiment scale, i.e. whether the method stay effective on SOTA models on AndroidWorld benchmark. Could the authors clarify on this?

---

> > > > ### Author Response · Authors · 2025-08-05
> > > > **Clarification on Applicability of GUI-Reflection to SOTA Models**
> > > >
> > > > Thank you for your thoughtful follow-up. We would like to clarify the applicability of our GUI-Reflection framework to SOTA models and its broader implications:
> > > >
> > > > - The final performance of GUI agents is influenced by multiple orthogonal factors such as model size, base model reasoning capabilities, and GUI-specific training data. Our GUI-Reflection framework focuses specifically on enhancing reflection and self-correction capabilities, which are **complementary and orthogonal** to the above factors. Our experiments demonstrate that, when controlling for other variables, injecting reflection-oriented design at each training stage yields consistent improvements. We also validated this across different model scales.
> > > >
> > > > - While recent SOTA models like Seed1.5-VL show strong overall performance, we found through **manual inspection and keyword-based analysis** (e.g., “mistake”, “incorrect”, “unsuccessful”) that their trajectories **rarely exhibit explicit reflective behaviors**. This suggests that even SOTA models **still lack robust self-correction abilities**, motivating the need for explicit injection of such capabilities.
> > > >
> > > > - Since models like UI-TARS and Seed1.5-VL have already gone through multiple GUI-related training stages (pretraining, SFT, RL), ideally, our GUI-Reflection framework should be integrated during their original training. However, due to the lack of access to their training data and training details, this is not feasible. Nevertheless, we are actively exploring applying  GUI-Reflection training on top of these existing SOTA models (e.g., UI-TARS-1.5, GLM-4.1V-Thinking), and early experiments are promising in terms of enhancing reflective behavior and downstream performance.
> > > >
> > > > We hope this clarifies that GUI-Reflection is designed to be model-agnostic and can enhance even the strongest existing models by targeting a complementary skill set currently underdeveloped in most GUI agents.

---

> > > > > ### Comment · Reviewer_DG2g · 2025-08-05
> > > > > **Feedback**
> > > > >
> > > > > Thank the author for the responses, I have raised my rating to 4 since authors address many of my concerns.
> > > > >
> > > > > That being said, I am still concerned about several points that need further experiments:
> > > > >
> > > > > 1. method robustness to model scale and base performance.
> > > > > 2. benchmark varieties. As of today, MLLM-as-a-judge can not be fully trusted on complex real-world task, and the nearly 100% judge success rate on the relatively simple AndroidWorld task largely contributes to the effectiveness of the model. I doubt if MLLM-as-a-judge can still work in this method for more difficult tasks.
> > > > >
> > > > > Addressing about concerns by empirical evidences could greatly strengthen the paper.

---

> > > > > > ### Author Response · Authors · 2025-08-06
> > > > > >
> > > > > > Thank you for the follow-up and the updated rating. We fully agree that further empirical studies would strengthen the work. We are currently expanding experiments to test robustness across model scales and base capabilities, as well as evaluating our framework on more complex GUI tasks beyond AndroidWorld with alternative judges. We will report these in the final version.

---

### Note · Authors · 2025-08-13

Dear AC, SAC, and Reviewers,

We sincerely thank you for the thoughtful reviews, constructive feedback, and active engagement during the discussion phase. Here, we summarize our acknowledged contributions and the discussion process.

**Recognized Contributions** We are grateful that reviewers acknowledged key contributions of our work:

- Addressing an important and timely problem for GUI agents by explicitly injecting reflection and error-correction abilities (all reviewers).

- The proposed solution is reasonable (reviewer pU3E), useful, reusable (reviewer phbt), and automatic (reviewer DG2g).

- Competitive performance on AndroidWorld benchmark (all reviewers).

**Rebuttal & Discussion Summary** In response to reviewers feedback, we have:

- Validated Stage Necessity: Ablations show that removing reflection designs from any stage reduces the end-to-end performance.

- Demonstrated Scalability: Added results on InternVL3-78B, improving AndroidWorld SR from 31.5% → 43.1%.

- Strengthened Comparisons: Iterative reflection tuning outperforms rejection sampling (+8.6) and teacher distillation (+12.1).

- Ensured Reliability: Audits confirm high accuracy in the MLLM-related verification and annotation process.

**Planned Final Version Additions**

- Integration of new large-scale results, stage-wise ablations, and alternative baseline comparisons

- Inclusion of audit tables about MLLM-related verification and annotation, and implementation details for reproducibility

- Expanded discussion on generalization to more challenging benchmarks and cross-platform applicability

We are encouraged that DG2g and phbt raised their scores to 4, with pU3E confirming that most concerns have been resolved. We look forward to incorporating these improvements in the final version and believe the strengthened evidence and broad applicability put our work in a strong position for acceptance.

Best regards,
Authors

---

### Decision · Program_Chairs · 2025-09-17

**Decision:**

Accept (poster)

**Comment:**

This paper presents GUI-Reflection, a framework engineered to empower multimodal GUI agents with self-reflection and error-correction functionalities, thereby addressing their prevalent inability to recover from errors. In contrast to conventional GUI models that rely solely on error-free training datasets and lack recovery mechanisms, GUI-Reflection integrates reflection behaviors through pre-training, offline supervised fine-tuning, and online reflection tuning. This comprehensive approach substantially enhances performance, enabling an 8B model to surpass prior models of its size and compete with significantly larger systems on the AndroidWorld benchmark.

The reviewers concur that the paper addresses a pertinent and timely issue: equipping GUI agents with the capability to recognize and recover from errors. The proposed method is demonstrated to yield robust and competitive outcomes on the public AndroidWorld benchmark. The major concern arises from the restricted experimental scope, which employs a restricted set of baseline models for comparison and testing primarily confined model scale and platform. Consequently, the reviewers suggest that the paper can benefit from a revision incorporating these modifications to comprehensively evaluate the paper within this revised context.

The authors are also engaged in the rebuttal phase, proactively answering questions and addressing concerns raised in the initial reviews.